# RAMAC: Multimodal Risk-Aware Offline Reinforcement Learning and the Role of Behavior Regularization

## Abstract

In safety-critical domains where online data collection is infeasible, offline reinforcement learning (RL) offers an attractive alternative but only if policies deliver high returns without incurring catastrophic lower-tail risk. Prior work on risk-averse offline RL achieves safety at the cost of value or model-based pessimism and restricted policy classes that limit policy expressiveness, whereas diffusion/flow-based expressive generative policies trained with a behavioral-cloning (BC) objective have been used only in risk-neutral settings. Here, we address this gap by introducing the **Risk-Aware Multimodal Actor-Critic (RAMAC)**, which couples an expressive generative actor with a distributional critic and, to our knowledge, is the first model-free approach that learns *risk-aware expressive generative policies*. RAMAC differentiates a composite objective that adds a Conditional Value-at-Risk (CVaR) term to a BC loss, achieving risk-sensitive learning in complex multimodal scenarios. Since out-of-distribution (OOD) actions are a major driver of catastrophic failures in offline RL, we further analyze OOD behavior under prior-anchored perturbation schemes from recent BC-regularized risk-averse offline RL. This clarifies why a behavior-regularized objective that directly constrains the expressive generative policy to the dataset support provides an effective, risk-agnostic mechanism for suppressing OOD actions in modern expressive policies. We instantiate RAMAC with a diffusion-based actor, using it both to illustrate the analysis in a 2-D risky bandit and to deploy OOD-action detectors on Stochastic-D4RL benchmarks, empirically validating our insights. Across these tasks, we observe consistent gains in $\mathrm{CVaR}_{0.1}$ while maintaining strong returns.

## 1 Introduction

In high-stakes applications such as autonomous driving, robotics, finance, and healthcare, where real-life explorations may lead to catastrophic consequences, offline RL offers a safe approach for generating policies that not only maximize long-horizon returns but also *tightly control risk* (Levine et al., 2020). Recent expressive generative policies (Wang et al., 2023; Park et al., 2025; Koirala & Fleming, 2025) can capture multimodal behavior and thus excel in achieving high expected return, yet their primary use has been limited to *risk-neutral* settings. Conversely, existing risk-averse algorithms ensure safety by enforcing conservatism or restricted policy classes that constrain policy expressiveness. (Kumar et al., 2020; Urpí et al., 2021; Ma et al., 2021). This paper asks: *Can we obtain safety without sacrificing expressiveness?*

We answer in the affirmative by proposing the **Risk-Aware Multimodal Actor-Critic (RAMAC)** framework (Fig. 1). RAMAC couples an expressive generative actor with a distributional critic and *differentiates a combination of BC and distributional risk (instantiated with CVaR) gradients through the generative process* (Di Castro et al., 2012; Chow et al., 2015). This unifies high expressiveness with robust tail-risk control while directly constraining the generative policy to the data support, addressing two central safety concerns in offline RL: catastrophic tail outcomes and out-of-distribution (OOD) actions.

Prior offline-RL approaches can be organized by mechanism: **(i) Policy regularization** constrains the policy to the data manifold via divergence minimization or policy priors, improving stability but often sacrificing policy expressiveness on complex tasks with risk-neutral examples such as (Fujimoto et al., 2019; Wu et al., 2019; Kumar et al., 2019; Fujimoto & Gu, 2021) and risk-aware methods with *prior-anchored perturbation* designs such as (Urpí et al., 2021; Chen et al., 2024b). **(ii) Value conservatism** reduces optimistic extrapolation, but can underestimate the value of infrequent yet high-return in-distribution modes due to global pessimism and data imbalance in both risk-neutral (Kumar et al., 2020) and risk-aware instances (Ma et al., 2021). **(iii) Model-based pessimism** bounds transition uncertainty with ensembles and penalties, at the cost of compounding model errors at scale again under both risk-neutral (Yu et al., 2020; 2021; Rigter et al., 2022) and risk-aware (Rigter et al., 2023) settings. **(iv) Expressive generative policies** faithfully clone multimodal behavior and achieve state-of-the-art mean returns, but limited use only in *risk-neutral* applications (Chen et al., 2021; Janner et al., 2022; Ajay et al., 2022; Wang et al., 2023; Hansen-Estruch et al., 2023; Park et al., 2025) including closest concurrent works pairing diffusion with distributional critics (Zhang et al., 2025; Liu et al., 2025). In these methods, expressive diffusion/flow policies trained with BC-style objectives have been widely adopted in offline RL.

Despite compelling results from these expressive generative policies in risk-neutral RL, their potential in offline risk-aware RL remains largely untapped. Among behavior-regularized risk-averse methods, the most prominent approaches that leverage expressive priors rely on *prior-anchored perturbation* (Urpí et al., 2021; Chen et al., 2024b), which trade away much of the multimodal capacity and, as we show theoretically, can still incur OOD actions.

Here, we aim to leverage the advantages of expressive policies without compromising risk-aversion or increasing the OOD action rate. To this end, inspired by the success of risk-neutral expressive policies such as (Wang et al., 2023), RAMAC optimizes a joint objective composed of a BC loss on the generative actor and a CVaR objective on the return distribution. Unlike prior diffusion/flow-based offline RL methods, which rely on BC-style objectives primarily as an empirical safeguard against extrapolation (Chen et al., 2024a), our BC term induces a forward-KL upper bound on the probability of taking off-support actions, directly linking behavior regularization and tail-risk control (Sec. 4). Combined with the CVaR-based risk term, this yields a principled behavior-regularized objective for expressive risk-aware policies. To our knowledge, this is the first objective-level characterization of how a BC-regularized expressive policy controls OOD behavior. We empirically show that RAMAC yields high expected return while minimizing risk on complex multimodal offline benchmarks.

Our contributions can be summarized as:

- **Risk-aware expressive policy learning:** We leverage expressive policies in the context of risk-aware RL by introducing RAMAC, a model-free framework for learning risk-aware expressive generative policies. Our primary instantiation is a diffusion-based actor, **RADAC**.

- **Theoretical insight:** We provide a geometric analysis of OOD behavior in offline RL, showing that prior-anchored perturbation schemes can still place probability mass on off-support actions even with expressive priors. In contrast, for expressive generative policies, we derive a forward-KL upper bound on the per-state OOD action probability, explaining why a behavior-regularized objective on the generative actor is an effective, risk-agnostic way to control OOD visitation.

- **Experimental evaluation:** On Stochastic-D4RL benchmarks, RADAC outperforms baselines on CVaR while maintaining competitive mean return on most tasks. We additionally report a flow-matching variant, **RAFMAC**, in App. E, which shows similar trends. We also (i) visualize our geometric analysis in a 2-D risky bandit that contrasts *risk-aware prior-anchored perturbation with expressive priors* and *risk-aware expressive generative policies*, and (ii) use OOD-action detectors on Stochastic-D4RL to empirically validate the theoretical insights.

## 2 PRELIMINARIES

**Offline RL** We consider a finite-horizon Markov Decision Process (MDP) $\mathcal{M} = (\mathcal{S}, \mathcal{A}, P, r, \gamma, H)$ with state space $\mathcal{S}$, action space $\mathcal{A}$, transition kernel $P(\cdot \mid s, a)$, reward function $r(s, a)$, discount factor $\gamma \in (0, 1)$, and horizon $H \in \mathbb{N}$. (Sutton et al., 1998). In offline RL, the learner is given only a static dataset $\mathcal{D} = \{(s_i, a_i, r_i, s_i')\}_{i=1}^N$ collected by some unknown behavior

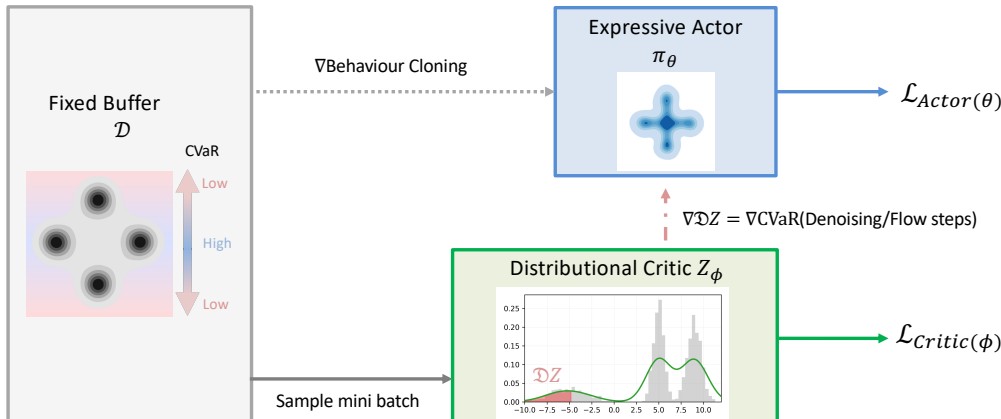

Figure 1: **RAMAC pipeline.** From the offline buffer $\mathcal{D}$ (gray), the distributional critic $Z_\phi$ (green) fits the return law with a quantile loss and aggregates its lower tail into a CVaR signal. That signal is differentiated through the generative path of the actor $\pi_\theta$ (blue; diffusion or flow), which is trained with the composite objective $\mathcal{L}_\pi = \mathcal{L}_{\mathrm{BC}} + \eta \mathcal{L}_{\mathrm{Risk}}$ to shift mass away from low-quantile regions while staying on-manifold.

policy $\beta$, and cannot further interact with the environment. (Prudencio et al., 2023). Let $\mathrm{supp}(\mathcal{D})$ denote the dataset's empirical state–action support. The objective is to learn a policy $\pi$ that maximizes the expected return $J(\pi) = \mathbb{E}_{\pi,P}[\sum_{t=0}^{H-1} \gamma^t r_t]$ without extra environment interaction. The central challenge is *distributional shift* (i.e., OOD): When $\pi$ visits $(s,a) \notin \mathrm{supp}(\mathcal{D})$, value estimates $Q(s,a)$ extrapolate and can become arbitrarily inaccurate (Kumar et al., 2020). Policies that place non-negligible mass on such OOD actions may therefore suffer catastrophic failures at deployment. Prior work alleviates this issue with behavior regularization, conservative critics, or model-based pessimism.

**Behavior–Regularized Actor–Critic (BRAC)**  A large family of offline methods uses an actor–critic with an explicit proximity term to the behavior policy (Nair et al., 2020; Fujimoto & Gu, 2021; Wu et al., 2019; Kumar et al., 2019). A representative actor–critic objective takes the form:

$$\mathcal{L}_{\mathrm{Actor}}(\theta) = \mathbb{E}_{s\sim\mathcal{D},\, a\sim\pi_\theta(\cdot|s)}\big[-Q_\phi(s,a) - \alpha \log \pi_\theta(a \mid s)\big], \qquad (1)$$

$$\mathcal{L}_{\mathrm{Critic}}(\phi) = \mathbb{E}_{(s,a,r,s')\sim\mathcal{D},\, a'\sim\pi_\theta(\cdot|s')}\big(Q_\phi(s,a) - [r + \gamma Q_{\bar\phi}(s',a')]\big)^2. \qquad (2)$$

Here the second term $-\alpha \log \pi_\theta(a \mid s)$ plays the role of a *behavior regularizer*: it is typically instantiated as a KL- or behavioral-cloning proximity term that keeps $\pi_\theta(\cdot \mid s)$ close to the behavior policy $\beta(\cdot \mid s)$. Empirically, BRAC-style objectives have turned out to be surprisingly strong in offline RL (Tarasov et al., 2023). In this work, we extend this behavior-regularized pattern to a *distributional* actor–critic in which the critic is expanded into *distributional critic* that evaluates a coherent risk measure (CVaR) in place of the mean $Q$.

**Distributional RL and Risk Measures**  Standard actor–critic methods including BRAC optimize the expected return by learning the mean action-value function $Q^\pi(s,a) = \mathbb{E}[Z^\pi(s,a)]$ as shown in Eq. 2. Distributional RL instead models the entire *return distribution* $Z^\pi(s,a)$ (Bellemare et al., 2017). The distributional Bellman operator is:

$$(\mathcal{T}^\pi Z)(s,a) \stackrel{d}{=} r(s,a) + \gamma Z(s',a'), \quad s' \sim P(\cdot|s,a),\ a' \sim \pi(\cdot|s'). \qquad (3)$$

A common parameterization uses an Implicit Quantile Network (IQN) (Dabney et al., 2018) to approximate the inverse CDF $Z_\phi(s,a;\tau) \approx F_{Z^\pi(s,a)}^{-1}(\tau)$ for quantile levels $\tau \in (0,1)$. Access to quantiles enables coherent risk measures $\mathcal{D}(\cdot)$ that emphasize different parts of the return distribution. In this work we focus on the Conditional Value-at-Risk (CVaR), a widely used instantiation of $\mathcal{D}(\cdot)$ as risk-averse objective. For a risk level $\alpha \in (0,1]$, the CVaR admits the integral form used for

actor gradients:

$$\text{CVaR}_\alpha(X) = \frac{1}{\alpha} \int_0^\alpha F_X^{-1}(\tau)\, d\tau. \tag{4}$$

Optimizing $\text{CVaR}_\alpha$ encourages policies that trade some mean performance for improved behavior in the worst $\alpha$-fraction of trajectories, which is crucial in safety-critical settings. In RAMAC, the distributional critic provides quantile estimates from which CVaR and its gradients with respect to actions can be computed and backpropagated through the policy.

**Expressive Generative Policies as Differentiable Trajectories**  Recent offline RL methods stay within the behavior-regularized actor-critic template of Eqs. 1 and 2, but replace the simple parametric actor with an expressive conditional generative model  (Wang et al., 2023; Kang et al., 2023; Park et al., 2025). Given a state $s$ and latent $z \sim \mathcal{N}(0, I)$, the policy generates an action $a = \psi_\theta(s, z)$ along a *differentiable path*  (Janner et al., 2022; Wang et al., 2023), while an explicit behavior term keeps $\psi_\theta$ close to the dataset actions signals. We focus on the two families:

(i) *Diffusion policies* model a reverse-time stochastic differential equation (SDE) over actions (Song et al., 2021b),

$$\mathrm{d}_t \mathbf{a}_t = f_\theta(t, \mathbf{a}_t, s)\, \mathrm{d}t + g(t)\, \mathrm{d}\mathbf{w}_t, \tag{5}$$

where a forward noising process gradually corrupts dataset actions into near-Gaussian noise, and the network $f_\theta$ learns to reverse this process conditioned on the state $s$.

(ii) *Flow-matching policies* solve a deterministic ODE (Lipman et al., 2023),

$$\frac{\mathrm{d}\mathbf{a}_t}{\mathrm{d}t} = v_\theta(t, \mathbf{a}_t, s). \tag{6}$$

where a neural vector field $v_\theta$ transports samples from a simple base distribution to the data distribution along a continuous trajectory. Integrating Eq. 6 from an initial noise sample yields an action conditioned on $s$.

In both cases, the overall map $\psi_\theta : (s, z) \mapsto a$ is fully differentiable, and the behavior term encourages $\psi_\theta$ to approximate the behavior action distribution itself; the critic then fine-tunes this expressive behavior model using scalar signals. Prior work typically uses expected-value or advantage-based signals from a mean-value critic (as in Eq. 2) to update the generative policy, yielding a risk-neutral generative BRAC method. In our case, a distributional critic instead provides explicit *distributional risk signals*, namely tail-sensitive values such as $\text{CVaR}_\alpha(Z^\pi(s, a))$ derived from the return distribution (as in Eq. 4), that shape the diffusion/flow policy under the same behavior-regularized template; the exact loss is introduced in Sec. 3.

## 3    Risk-Aware Multimodal Actor-Critic (RAMAC)

We now introduce the **Risk-Aware Multimodal Actor-Critic (RAMAC)**. At its core, RAMAC operates in two stages: First, a distributional critic parameterized as an IQN learns the full conditional distribution of returns. Second, a generative actor, instantiated as either a diffusion policy or a flow-matching policy, is guided *jointly* by two terms in the objective function: (i) BC term that constrains the policy to the data manifold and (ii) CVaR term extracted from the critic's lower tail. The latter pushes the probability mass away from low-probability, catastrophic regions and preserves the high-reward modes (Fig. 2). The former acts as a regularizer: by keeping $\pi_\theta(\cdot \mid s)$ close to the empirical behavior distribution, it limits OOD visitation, mirroring the BRAC-style regularizers in Eq. 1 and connecting directly to our forward-KL OOD bound in Sec. 4.

### 3.1    Distributional Critic

Risk-sensitive objectives such as CVaR require access to the entire return distribution. We therefore adopt a distributional critic $Z_\phi$ via IQN (Dabney et al., 2018), building on the distributional Bellman operator in Eq. 3 Bellemare et al. (2017). We minimize a distributional Bellman residual with a quantile-Huber loss (with $\kappa = 1$):

$$\mathcal{L}_{\text{critic}}(\phi) = \mathbb{E}_{(s,a,r,s') \sim \mathcal{D},\, a' \sim \pi_\theta(\cdot|s'),\, \tau,\tau' \sim \mathcal{U}(0,1)} \Big[ \mathcal{L}_\kappa \big( r + \gamma Z_{\bar{\phi}}(s', a'; \tau') - Z_\phi(s, a; \tau)\, ; \tau \big) \Big]. \tag{7}$$

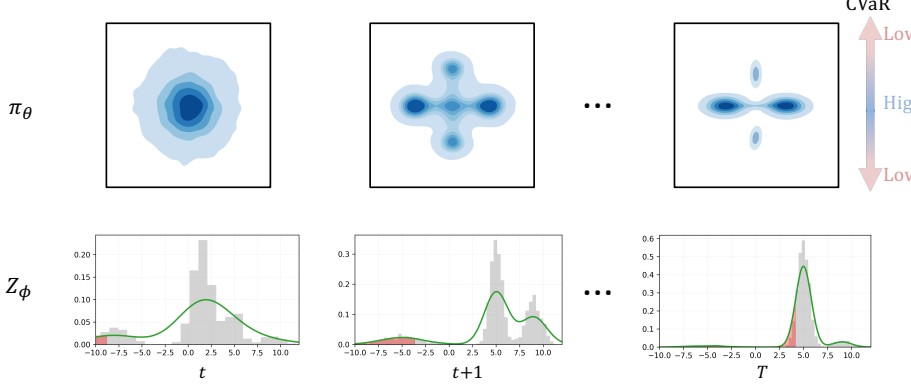

Figure 2: **RAMAC learning dynamics (conceptual).** *Top:* policy density $\pi_\theta(a \mid s)$ induced by the reparameterized actor $a = \psi_\theta(s, z)$ *(Eq. 8)* over training. *Bottom:* critic return distribution $Z_\phi(s, a, \tau)$ with low quantiles highlighted (red); the actor is updated by the CVaR objective *(Eqs. 9–12)* while the critic is trained via the IQN loss *(Eq. 7)*. CVaR updates steer mass away from low-quantile regions while preserving multimodal high-reward modes.

This yields calibrated lower-tail quantiles that will directly drive the risk-aware actor update in Sec. 3.2.

### 3.2 RISK-AWARE GENERATIVE ACTOR

An action is sampled as:
$$a = \psi_\theta(s, z), \qquad z \sim \mathcal{N}(0, I). \tag{8}$$

We define CVaR at level $\alpha$ through the critic's quantiles and use a Monte Carlo estimator:

$$\mathrm{CVaR}_\alpha\big(Z_\phi(s, a)\big) = \frac{1}{\alpha} \int_0^\alpha Z_\phi(s, a; \tau)\, d\tau \approx \frac{1}{K} \sum_{k=1}^{K} Z_\phi\big(s, a; \tau_k\big), \quad \tau_k \sim \mathcal{U}(0, \alpha). \tag{9}$$

The risk loss maximizes this quantity. This is equivalent to minimizing the negative CVaR [1]:

$$\mathcal{L}_{\mathrm{Risk}}(\theta) = - \mathbb{E}_{s \sim \mathcal{D},\, a \sim \pi_\theta(\cdot|s)}\big[\mathrm{CVaR}_\alpha\big(Z_\phi(s, a)\big)\big]. \tag{10}$$

### 3.3 BEHAVIOR-REGULARIZED OBJECTIVE

The complete policy objective balances risk aversion with fidelity to the offline dataset. We instantiate fidelity through a standard behavior cloning (BC) term that encourages the policy to reproduce the behavior distribution. We define:

$$\mathcal{L}_{\mathrm{BC}}(\theta) = - \mathbb{E}_{(s,a) \sim \mathcal{D}}\big[\log \pi_\theta(a \mid s)\big], \tag{11}$$

which matches the BRAC-style behavior regularizer in Eq. 1 up to a scaling of the coefficient.[2] It combines the risk term with a standard behavior cloning (BC) loss, $\mathcal{L}_{\mathrm{BC}}(\theta)$:

$$\mathcal{L}_\pi(\theta) = \underbrace{\mathcal{L}_{\mathrm{BC}}(\theta)}_{\text{data fidelity}} + \eta \underbrace{\mathcal{L}_{\mathrm{Risk}}(\theta)}_{\text{tail-risk aversion}}. \tag{12}$$

where $\eta$ is a hyperparameter. Our primary instantiation is a diffusion policy (**RADAC**), while an additional flow-matching variant (**RAFMAC**) is reported in the App. E. We show a pseudocode for RAMAC in Algorithm 1 and describe the full implementation details in App. D

---

[1] This specific loss, instantiated with CVaR, is what we refer to as $\mathcal{L}_{\mathrm{CVaR}}$ in our architectural diagrams for clarity.

[2] For diffusion/flow actors, we implement $\mathcal{L}_{\mathrm{BC}}$ via standard denoising / flow-matching losses.

---

**Algorithm 1 Risk-Aware Multimodal Actor-Critic (RAMAC)**

---

1: **Initialize** policy network $\pi_\theta$, critic $Z_\phi$, target critic $Z_{\bar\phi}$; mini–batch size $B$, risk level $\alpha$, critic–tail samples $K$, Exponential Moving Average (EMA) rate $\rho$.
2: **repeat**
3:     Sample a mini–batch $\{(s,a,r,s')\}_{b=1}^B \sim \mathcal{D}$.
4:     **Training Critic:**
5:     Sample $z' \sim \mathcal{N}(0,I)$ and set $a' = \psi_\theta(s',z')$ *(Eq. 8)*;
6:     Sample $\tau,\tau' \sim \mathcal{U}(0,1)$ and update $\phi$ by minimizing $\mathcal{L}_{\text{critic}}(\phi)$ *(Eq. 7)*.
7:     **Training Actor:**
8:     Sample $z \sim \mathcal{N}(0,I)$ and set $a = \psi_\theta(s,z)$ *(Eq. 8)*;
9:     Sample $\tau_1,\ldots,\tau_K \sim \mathcal{U}(0,\alpha)$ and update $\theta$ by minimizing $\mathcal{L}_\pi(\theta)$ *(Eq. 12)*.
10:     **Target update:** $\bar\phi \leftarrow \rho\,\bar\phi + (1-\rho)\,\phi$.
11: **until** converged

---

## 4 BEHAVIOR REGULARIZATION IN OFFLINE RL

Prior work has demonstrated the importance of behavior regularization in offline RL due to its ability to constrain the learned policy to the data manifold, curb value extrapolation, and stabilize improvement (Tarasov et al., 2023). A commonly adopted regularization scheme in offline risk-aware RL is prior-anchored perturbation method (e.g. ORAAC, UDAC)[3] (Urpí et al., 2021; Chen et al., 2024b), which uses a linear mixing of a pretrained BC policy with the RL actor (perturbation). Here, we first discuss the limitation of this regularization approach. We then demonstrate the advantages of our adopted scheme, namely, behavior-regularized objective method (shown in Eq. 12).

### 4.1 PRIOR-ANCHORED PERTURBATION AND ITS LIMITATIONS

In this approach, policy output can be written as:

$$a \;=\; b \;+\; \zeta_\psi(s,b), \qquad b \sim G_\phi(\cdot \mid s), \;\; \|\zeta_\psi(s,b)\| \le \Phi, \tag{13}$$

where $\zeta_\psi$ is a *learned residual* (optimized to increase $Q$ or CVaR) and the norm bound $\Phi$ *keeps updates close to the anchor*. Define the anchor support $\mathcal{S}_G(s)$ (the region in action space where $G_\phi(\cdot \mid s)$ places mass), the full action space $\mathbb{R}^d$, and the $\Phi$-radius ball of $b$

$$B_\Phi(b) \;=\; \{\, a \in \mathbb{R}^d : \|a-b\|_2 \le \Phi \,\}.$$

where any perturbed action $a = b+\zeta_\psi$ with $\|\zeta_\psi\| \le \Phi$ lies in $B_\Phi(b)$. Hence on-manifold deployment is guaranteed by the *safety margin* condition

$$\text{dist}\big(b,\, \mathbb{R}^d \backslash \mathcal{S}_G(s)\big) \;>\; \Phi \quad\Longrightarrow\quad B_\Phi(b) \subseteq \mathcal{S}_G(s) \;\text{ and }\; a \in \mathcal{S}_G(s) \;\text{ for all } \|\zeta_\psi\| \le \Phi,$$

where $\text{dist}(x,A) := \inf_{y \in A} \|x-y\|_2$ denotes Euclidean distance. OOD can still occur when this margin fails. This method provides a convenient *local* improvement rule, however, prior work has observed it suffers from *poor mode coverage* in multimodal action spaces (Wang et al., 2023). In addition to the identified limitations, we show *distinct geometric weakness* that can occur even without multimodality; having multiple modes merely magnifies the effect.

**Lemma 4.1.** Fix $s$ and write $I_s = \mathcal{S}_G(s)$ and $O_s = \mathbb{R}^d \setminus I_s$. Suppose there exist an anchor $b^\star \in I_s$ and a radius $\Phi > 0$ such that $\lambda\big(B_\Phi(b^\star) \cap O_s\big) > 0$, and the policy $\pi_{\text{anch}}(\cdot \mid s)$ induced by Eq. 13 admits a density $p(\cdot \mid s)$ with

$$p(a \mid s) \;\ge\; c > 0 \quad \text{for all } a \in B_\Phi(b^\star).$$

Then its per-state OOD probability

$$\delta_s(\pi_{\text{anch}}) \;:=\; \pi_{\text{anch}}(O_s \mid s)$$

satisfies

$$\delta_s(\pi_{\text{anch}}) \;\ge\; c \cdot \lambda\big(B_\Phi(b^\star) \cap O_s\big) \;>\; 0.$$

In particular, as long as the density on $B_\Phi(b^\star)$ remains bounded below by $c > 0$, further training of the residual cannot drive $\delta_s(\pi_{\text{anch}})$ to zero. Proof appears in App. B.1.

---

[3]For simplicity and consistency with our experiments, we will refer to UDAC as *ORAAC–Diffusion*

Lemma 4.1 captures how the geometry of the anchor support forces a strictly positive OOD probability once the anchor ball overlaps the complement $O_s$. Thin or nonconvex supports make such low-margin anchors $b^\star$ unavoidable, and gradients that are not constrained to lie tangent to the data manifold inevitably push some mass across the boundary (see App. B.1 for a more detailed discussion).

## 4.2 Behavior-Regularized Objective: Why it works better

In contrast to prior-anchored perturbation, as shown in Eq. 12, the BC term is applied *directly* to the *deployed* generative policy $\pi_\theta$. For explicit-likelihood actors, the BC loss is the Negative Log-Likelihood (NLL): $\mathbb{E}_{(s,a)\sim\mathcal{D}}[-\log \pi_\theta(a \mid s)] = H(\beta(\cdot \mid s)) + D_{\mathrm{KL}}\big(\beta(\cdot \mid s) \,\|\, \pi_\theta(\cdot \mid s)\big)$, so minimizing NLL shrinks the forward KL up to the data-dependent constant $H(\beta)$. We therefore monitor the BC loss as a practical proxy for $D_{\mathrm{KL}}(\beta\|\pi_\theta)$.

**Proposition 1.** For each state $s$, let $I_s = \{a : \beta(a \mid s) > 0\}$ and $O_s = I_s^c$. Assume $\beta \ll \pi_\theta$ (absolute continuity on $I_s$). Then the per-state OOD probability $\delta_s(\pi_\theta) := \pi_\theta(O_s \mid s)$ satisfies

$$\delta_s(\pi_\theta) \leq 1 - \exp\Big\{ - D_{\mathrm{KL}}\big(\beta(\cdot \mid s) \,\|\, \pi_\theta(\cdot \mid s)\big)\Big\}. \tag{14}$$

This shows that shrinking forward KL via BC can suppress per–state OOD, controlled by the selection of $\eta$ (Eq. 12), hence, avoiding an important challenge of offline RL discussed in Sec. 2 (Proof appears in App. B.2).

We can expand this risk-agnostic mechanism to bound its impact on tail risk. Let $\delta_s(\pi_\theta)$ be the OOD mass at state $s$. Under bounded rewards $Z_{\min} \leq r \leq Z_{\max}$, a simple mixture argument shows that for any risk level $\alpha \in (0, 1]$,

$$\mathrm{CVaR}_\alpha[X_s^\pi] \geq \mathrm{CVaR}_\alpha[X_{I_s}] - \frac{\delta_s(\pi_\theta)}{\alpha}(Z_{\max} - Z_{\min}),$$

where $X_{I_s}$ is the return restricted to the in-support region $I_s$. Thus, as long as $\delta_s(\pi_\theta)$ remains small, the worst-case CVaR degradation due to OOD mixing is only linear in the OOD mass. Combined with Proposition 1, this gives a simple CVaR degradation bound controlled by the forward KL in RAMAC.

## 4.3 Example

We design a 2-D contextual bandit with two disjoint modes (*Toy Risky Bandit*) to make the above geometric analysis concrete: Top left in Fig. 3 shows a ground truth that consists of *safe center* (moderate reward, no catastrophic tail) and a *risky ring* (higher mean with rare large penalties). The task isolates multimodality and lower-tail hazards. Below we introduce our baselines.

### 4.3.1 Expressive but risk-neutral controls

As a common notation, let $G_\phi(\cdot \mid s)$ denote an expressive conditional generator (diffusion model, flow, or conditional VAE) that induces the policy $a \sim G_\phi(\cdot \mid s)$. Our risk-neutral expressive baselines, DiffusionQL (Wang et al., 2023), FlowQL (Park et al., 2025), and a conditional VAE-QL(CVAE-QL), all minimize

$$\mathcal{L}_{\mathrm{RN}}(\phi) = \lambda_{\mathrm{BC}} \, \mathbb{E}_{(s,a)\sim\mathcal{D}}\big[\ell\big(a, G_\phi(s)\big)\big] - \mathbb{E}_{s\sim\mathcal{D},\, a\sim G_\phi(\cdot|s)}\big[Q_\psi(s,a)\big], \tag{15}$$

i.e., a standard BC loss plus a risk-neutral value-improvement term on top of an expressive actor $G_\phi$ with scalar critic $Q_\psi$. Each method simply instantiates $G_\phi$ (diffusion, flow, or CVAE) and its optimization hyperparameters; full details are given in App. F.1.

### 4.3.2 Prior-anchored perturbation (risk-aware)

As risk-aware anchor–perturbation baselines we use ORAAC, ORAAC–Diffusion, and ORAAC–Flow (Urpí et al., 2021; Chen et al., 2024b), all instantiated via Eq. 13 with their original coherent risk objectives. Concretely, given a behavior anchor $b \sim \beta(\cdot \mid s)$, each method learns a residual perturbation $f_\phi$ as in Sec. 4 and optimizes its own risk functional (e.g., CVaR or distorted expectations) under this prior-anchored geometry.

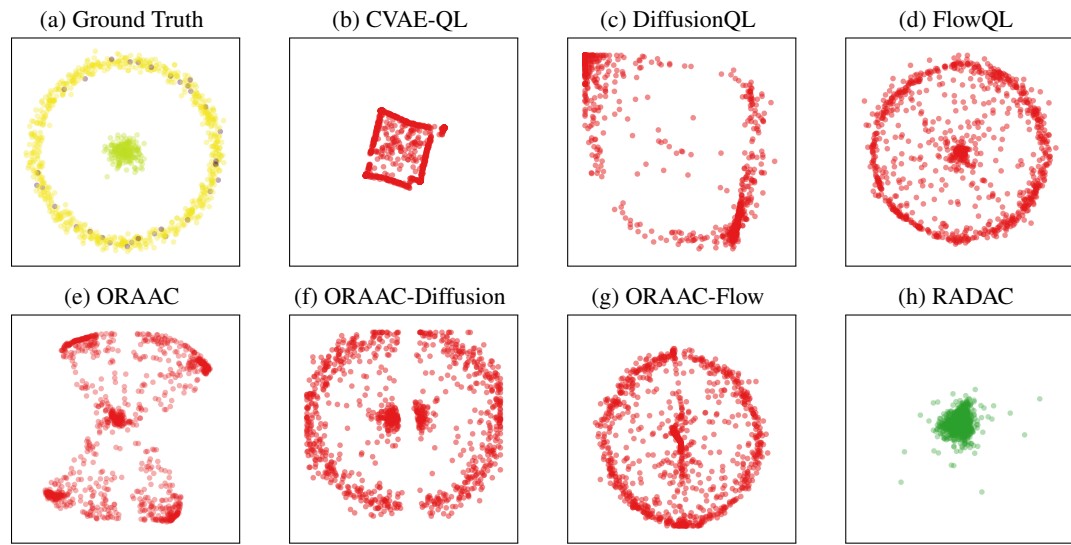

Figure 3: **Toy Risky Bandit Results** *Top:* Ground Truth consists of a safe center mode yellow-green and a risky ring where high-reward samples yellow are interspersed with catastrophic penalties (purple). Risk-neutral generative baselines concentrate on the risky ring or collapses topology. *Bottom:* Prior-anchored perturbation methods produce samples in the low-density inter-mode region, exhibiting OOD leakage. RADAC concentrates near the safe center without losing multimodality. See App. E for more results.

### 4.3.3 EXAMPLE RESULTS

The resulting policy distributions for various methods are shown in Fig. 3.

Risk-neutral expressive controls (Fig. 3 b-d): Overall, as expected, these methods are risk-blind and they chase high-$Q$ ridges without regard to the lower tail. FlowQL often preserves both modes but does not suppress mass on the hazardous ring; Diffusion QL drifts toward sparsely covered high-$Q$ pockets on the ring, yielding risk exposure; the CVAE variant collapses topology and fills low-density bridges.

Prior-anchored perturbation (Fig. 3 e-g): ORAAC and its diffusion/flow variants place substantial mass in the inter-mode low-density region, regardless of whether bc prior is expressive or not.

RADAC (Fig. 3 h): by sending CVaR signals from a distributional critic through the diffusion/flow trajectory while regularizing with BC, RADAC concentrates probability near the safe center without filling the gap. Full configuration and additional plots are in App. F.1. Overall, these patterns empirically support our theoretical analysis.

## 5 EXPERIMENTS

In this section, we evaluate **RADAC** on the Stochastic-D4RL benchmarks to validate both *risk awareness* and *policy expressiveness*. We also quantify the OOD action rate $\varepsilon_{\mathrm{act}}$ (Sec. 5.3) to link practice to the theoretical analysis in Sec. 4. Additional results, including the flow-matching instantiation **RAFMAC**, appear in App. E.

**Tasks** We augment standard D4RL locomotion tasks (Fu et al., 2020) with rare heavy-tailed penalties tied to forward velocity (HalfCheetah) or torso pitch angles (Hopper, Walker2d), together with early termination when the torso leaves a healthy range, following (Urpí et al., 2021). We evaluate on HOPPER, WALKER2D, and HALFCHEETAH using the MEDIUM−EXPERT and MEDIUM−REPLAY datasets, which are multimodal by construction. It lets us examine whether RAMAC learns *risk-aware* policies *without sacrificing multimodality*. During training, all methods receive per-transition rewards relabeled with the same stochastic hazard model that is also used at evaluation

Table 1: Stochastic–D4RL results over 5 seeds. We report Mean and $\text{CVaR}_{0.1}$; best in dark/ second in light shaded. The Full results with s.e. appear in App. E.2.

| Dataset | Metric | CQL | CODAC | ORAAC | FlowQL | DiffusionQL | RADAC |
|---|---|---|---|---|---|---|---|
| HalfCheetah-medium-expert | Mean | −66.66 | −0.12 | 796.06 | 844.14 | −20.71 | **916.64** |
| | CVaR | −135.39 | −0.11 | 742.94 | 754.44 | −76.39 | **805.25** |
| Walker2d-medium-expert | Mean | −21.52 | 23.96 | 969.62 | 1309.48 | −32.38 | **1708.68** |
| | CVaR | −64.88 | −43.88 | 358.55 | 468.15 | −116.19 | **573.22** |
| Hopper-medium-expert | Mean | −25.87 | 26.59 | **714.15** | 341.16 | −279.97 | 130.74 |
| | CVaR | −111.37 | −150.92 | **374.63** | −8.80 | −872.95 | −167.29 |
| HalfCheetah-medium-replay | Mean | −66.21 | −0.11 | 18.99 | 434.33 | 279.95 | **525.84** |
| | CVaR | −127.09 | −1.47 | −34.09 | 224.73 | 79.93 | **278.65** |
| Walker2d-medium-replay | Mean | −16.90 | 33.59 | 126.94 | 411.36 | 96.88 | **615.94** |
| | CVaR | −51.49 | −52.63 | −203.64 | 5.08 | 48.14 | **145.21** |
| Hopper-medium-replay | Mean | −16.25 | −47.83 | −18.00 | 373.16 | −2.79 | **385.58** |
| | CVaR | −118.70 | −160.08 | −129.25 | −62.24 | −51.33 | **−8.16** |

time, so they observe identical logged return signals and never see the underlying hazard mask or triggers directly; full construction details and per-task parameters appear in App. F.

**Baselines**   We compare against representative offline-RL methods covering value conservatism, distributional conservatism, anchor-perturb risk aversion, and risk-neutral expressive generators: CQL (Kumar et al., 2020), CODAC (Ma et al., 2021), ORAAC (Urpí et al., 2021), DiffusionQL (Wang et al., 2023), and FlowQL (Park et al., 2025). We defer detailed configurations to App. F.3.

**Evaluation**   Following protocols as those adopted in (Urpí et al., 2021; Wang et al., 2023), we train for 2000 epochs, each with 1000 gradient steps. We evaluate methods at fixed intervals of gradient steps and report (i) raw returns averaged over 5 seeds and (ii) episodic $\text{CVaR}_{0.1}$ computed over 50 rollouts in total (10 evaluation episodes per seed) to avoid normalization bias on the stochastic variants. For ORAAC and CODAC, we adopt the authors' risk-aware objectives (risk level $\alpha=0.1$ unless noted). For the other baselines, we tune hyperparameters within the same training budget to ensure fairness and otherwise use authors' recommended settings. Further protocol details, including the choice of quantile particles $N$ and tail-sample budget $K$, appear in App. E.6. Runtime and inference-latency comparisons for RADAC with DiffusionQL/FlowQL are reported in App. E.5.

## 5.1 RESULTS AND ANALYSIS

Table 1 reports Mean and $\text{CVaR}_{0.1}$ for RADAC alongside baselines. Across six tasks, RADAC delivers *strong lower tails with competitive or higher means*. FlowQL is often the strongest risk-neutral baseline, suggesting that flow-based generators can stabilize training even without explicit tail-risk shaping, but its lower-tail returns remain below RADAC. ORAAC regularizes toward a behavior anchor. It reliably handles sharp hazardous thresholds on such as HOPPER-MEDIUM-EXPERT but may fail to exploit high-reward modes, and can place mass in low-density between-mode regions.

## 5.2 QUALITATIVE SAFETY ANALYSIS

We visualize a contrast among three representative methods: risk-aware expressive generator RADAC, risk-neutral expressive generator DiffusionQL, and the anchor-perturb risk-averse method ORAAC. Fig. 4 plots the monitored distribution of policies against safe regions. RADAC concentrate probability mass inside or near the risk-free boundary while *actively reallocating probability onto high-return modes that lie within these regions*. DiffusionQL is tightly concentrated around zero because rare, high penalties depress bootstrapped values near the risk-free boundary. On the other hand, ORAAC regularizes toward a behavior anchor and thus settles at a low density between-mode when the anchor lies in risky-region. These features can be further observed in App. E.2, where we show empirical return distributions and risk–return frontiers.

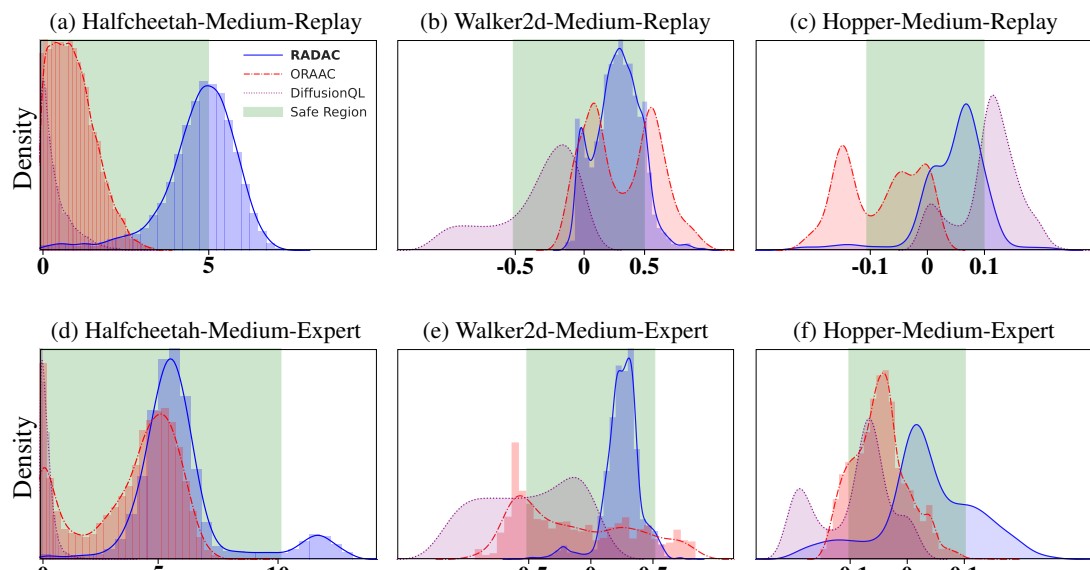

Figure 4: Policy distributions for RADAC, ORAAC, and DiffusionQL; shaded bands indicate safe operational ranges (HalfCheetah: $v \le 10$ for M-E, $v \le 5$ for M-R; Hopper: $|\theta| \le 0.1$; Walker2d: $|\theta| \le 0.5$). RADAC reduces mass beyond thresholds.

Table 2: OOD action rate ($\% \pm$ s.e.) on MEDIUM−EXPERT. $\kappa=3$).

| Task | RADAC (ours) | ORAAC |
|------|--------------|-------|
| HalfCheetah | $2.04 \pm 0.80$ | $6.15 \pm 1.5$ |
| Walker2d | $0.75 \pm 0.54$ | $10.84 \pm 1.98$ |
| Hopper | $0.77 \pm 0.56$ | $2.68 \pm 1.01$ |

### 5.3 EMPIRICAL ANALYSIS OF THEORETICAL INSIGHTS

We now provide measurements of OOD actions to validate the insights in Sec. 4. For each policy, we report $\varepsilon_{\text{act}}$, the fraction of evaluation actions whose 1-NN distance to the dataset exceeds $\kappa \times \text{median} \, d_{\text{NN}}$. Sec. 5.2 predicts that (i) (expressive, BC-regularized CVaR objective) should reweight probability *within* the data manifold, yielding low $\varepsilon_{\text{act}}$. (ii) ORAAC, being *less expressive* and based on anchor–perturbation, should exhibit *higher* $\varepsilon_{\text{act}}$ than RADAC; Table 2 confirms this prediction: RADAC retains low-OOD across tasks, consistent with BC-regularized OOD suppression; ORAAC is consistently higher than RADAC, matching the expected geometric leakage from anchor–perturbation, consistent with Sec. 4. Similar RADAC < ORAAC rankings hold under alternative OOD detectors (App. E.4). Overall, RADAC achieves risk awareness and expressiveness simultaneously while maintaining low OOD rates.

## 6 CONCLUSIONS

This paper introduces RAMAC, a model-free framework for *risk-aware* offline RL using *expressive* generative policies. This is done by coupling a distributional critic with an expressive generative actor and a simple BC+CVaR objective, instantiated in our main experiments as the diffusion-based RADAC with a flow-based variant in the appendix. Our analysis shows that applying BC directly to the deployed actor yields a forward-KL upper bound on per-state OOD mass and its contribution to CVaR degradation, and that prior-anchored perturbation can still leak into low-density regions on thin or non-convex supports. On stochastic-D4RL benchmarks with rare hazards, RADAC achieves consistently stronger tail returns (CVaR$_{0.1}$) with competitive or higher means and lower measured OOD rates than recent SOTA baselines in risk-aware/neutral offline RL.

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

## A    LIMITATIONS AND FUTURE DIRECTIONS

Practical remedies include adding a distilled *one-step* RL head to avoid recursive backprop and score/energy–weighted objectives that reduce full-path backprop for diffusion policies (Park et al., 2025; Ma et al., 2025) can be a possible future improvement. On the critic side, modeling the *return distribution* itself with diffusion/flow value networks is a natural extension that may improve tail calibration without sacrificing actor expressiveness (Liu et al., 2025; Agrawalla et al., 2025). Our theory is deliberately scoped to proposition-level guidance (e.g., KL–OOD relations) and does not provide finite-sample, high-probability guarantees under function approximation or partial coverage. Strengthening these to non-asymptotic guarantees, exploring alternatives to BC ( $f$-divergences or

Wasserstein) to trade off mode-covering vs. mode-seeking; extending analysis to *dynamic/spectral risk* and partial observability, and developing *risk-aware offline-to-online fine-tuning* that preserves on-manifold exploration are all promising directions for future work. In parallel, our empirical study does not yet explore high-confidence policy-improvement schemes (e.g., SPIBB-style constraints) in continuous control tasks; adapting these discrete/tabular or value-conservative methods to expressive generative actors is a complementary avenue for future work.

# B   PROOFS

## B.1   PROOF OF LEMMA 4.1

*Proof.* By assumption, $\pi_{\mathrm{anch}}(\cdot \mid s)$ has a density $p(a \mid s)$ on $B_\Phi(b^\star)$ with $p(a \mid s) \geq c > 0$. Thus

$$\delta_s(\pi_{\mathrm{anch}}) = \int_{O_s} p(a \mid s)\, da \;\geq\; \int_{B_\Phi(b^\star) \cap O_s} p(a \mid s)\, da \;\geq\; c \cdot \lambda\big(B_\Phi(b^\star) \cap O_s\big) > 0.$$

The final claim follows as long as the density lower bound $c$ on $B_\Phi(b^\star)$ is maintained during training. $\square$

### B.1.1   GEOMETRIC INTUITION BEHIND LEMMA 4.1

Lemma 4.1 formalizes the geometric intuition that when the anchor point $b$ lies near a thin or non-convex region of the support $\mathcal{B}_s$, any perturbation policy constrained within a ball centered at $b$ will unavoidably place some mass outside the dataset support $\mathcal{D}_s$. The following points provide an intuitive view:

- **Thin support near $b$:** When the local margin $m(b)$ is small, any ball $B_\Phi(b)$ with $\Phi \gtrsim m(b)$ necessarily overlaps $O_s$, so some residual updates produce OOD actions even if $\|\zeta_\psi\| \leq \Phi$.

- **Nonconvex support:** If $\mathcal{S}_G(s)$ is nonconvex (e.g., a ring/annulus), anchors can sit near holes or concavities, again yielding small $m(b)$ and forcing overlap between $B_\Phi(b)$ and $O_s$.

- **Gradients pushing off the data surface:** The residual $\zeta_\psi(s, b)$ is trained to increase $Q$ or CVaR and is not constrained to remain tangent to the manifold, so gradients can push along the normal direction toward the boundary of $B_\Phi(b)$, further amplifying leakage when $m(b) \leq \Phi$.

Even when a coherent risk objective (e.g., CVaR) is used to train $\zeta_\psi$, Lemma 4.1 implies that a nontrivial fraction of actions remains OOD whenever the behavior support is thin or nonconvex.

## B.2   PROOF OF PROPOSITION 1

*Proof.* Fix a state $s$ and define the $\beta$–support $I_s := \{a : \beta(a \mid s) > 0\}$ and its complement $O_s := I_s^c$. Assume $\beta \ll \pi_\theta$ on $I_s$ (so $\pi_\theta(I_s \mid s) > 0$ and the forward KL is finite).

Since $\beta(\cdot \mid s)$ has all its mass on $I_s$,

$$D_{\mathrm{KL}}(\beta \| \pi_\theta) = \int_{\mathcal{A}} \beta(a \mid s) \log \frac{\beta(a \mid s)}{\pi_\theta(a \mid s)}\, da \;=\; \int_{I_s} \beta(a \mid s) \log \frac{\beta(a \mid s)}{\pi_\theta(a \mid s)}\, da. \tag{1}$$

Define the normalization of $\pi_\theta$ to $I_s$:

$$\pi_I(a \mid s) \;:=\; \pi_\theta(a \mid s,\, a \in I_s) \;=\; \frac{\pi_\theta(a \mid s)}{\pi_\theta(I_s \mid s)}, \qquad a \in I_s,$$

so that on $I_s$ we have the identity $\pi_\theta(a \mid s) = \pi_\theta(I_s \mid s)\, \pi_I(a \mid s)$.

Substitute the above factorization into (1) and use $\log(xy) = \log x + \log y$:

$$\log \frac{\beta(a \mid s)}{\pi_\theta(a \mid s)} = \log \frac{\beta(a \mid s)}{\pi_\theta(I_s \mid s)\, \pi_I(a \mid s)} = \log \frac{\beta(a \mid s)}{\pi_I(a \mid s)} - \log \pi_\theta(I_s \mid s). \tag{2}$$

Plug (2) into (1) and split the integral:

$$D_{\text{KL}}(\beta\|\pi_\theta) = \int_{I_s} \beta(a \mid s) \log \frac{\beta(a \mid s)}{\pi_I(a \mid s)} \, da \; - \; \int_{I_s} \beta(a \mid s) \log \pi_\theta(I_s \mid s) \, da$$

$$= \underbrace{D_{\text{KL}}\big(\beta(\cdot \mid s) \, \| \, \pi_I(\cdot \mid s)\big)}_{\geq 0} \; - \; \log \pi_\theta(I_s \mid s) \underbrace{\int_{I_s} \beta(a \mid s) \, da}_{=1}. \qquad (3)$$

Here the last equality uses that $\log \pi_\theta(I_s \mid s)$ is constant in $a$, and that $\beta$ places total mass 1 on $I_s$.

From (3) and nonnegativity of KL,

$$D_{\text{KL}}(\beta\|\pi_\theta) \; \geq \; - \log \pi_\theta(I_s \mid s).$$

Exponentiating both sides gives

$$e^{-D_{\text{KL}}(\beta\|\pi_\theta)} \; \leq \; \pi_\theta(I_s \mid s).$$

Since $\pi_\theta(I_s \mid s) = 1 - \delta_s(\pi_\theta)$ with $\delta_s(\pi_\theta) := \pi_\theta(O_s \mid s)$, we obtain the per–state OOD bound

$$\delta_s(\pi_\theta) \; \leq \; 1 - \exp\big\{-D_{\text{KL}}(\beta(\cdot \mid s)\|\pi_\theta(\cdot \mid s))\big\}. \qquad \square$$

## C  RELATED WORKS

We review works most relevant to our *risk-aware generative trajectory* view—policies that map noise to actions through a differentiable path and how safety is enforced therein—while avoiding repetition of the core background already covered in the main text. For a broad taxonomy of offline RL, see Prudencio et al. (2023).

**Expressive generative policies**  The main paper reviews diffusion and flow-matching policies (e.g., DiffusionQL, Diffuser, IDQL, FlowQL). Here we note complementary developments not detailed there: (i) *DDIM-style imitation learning* that accelerates inference while keeping diffusion's expressiveness (Song et al., 2021a); (ii) *real-robot deployments* of diffusion policies demonstrating hardware viability (Chi et al., 2023); and on the flow side . These works bolster the case for expressive, differentiable policies but remain *risk-neutral* in objective design.

**Autoregressive generative baselines**  *Trajectory Transformer* (Janner et al., 2021) provides strong risk-neutral baselines by modeling returns/actions autoregressively; diffusion has also been used for open-loop planning (Chen et al., 2023). Because decoding is single-shot, these approaches lack a continuous generative path through which tail-risk gradients can be injected, leading to high mean performance without explicit lower-tail control.

**Risk-sensitive RL**  Beyond expectation-oriented objectives, risk-sensitive control formalizes tail-aware criteria via coherent/dynamic risk measures for MDPs (Ruszczyński, 2010). Among coherent measures, CVaR admits sampling- and policy-gradient formulations suitable for RL (Tamar et al., 2015b;a), and has been linked to robustness via CVaR–robust trade-offs (Chow et al., 2015). In the *offline* regime, safety is often operationalized as high-confidence off-policy evaluation/improvement from fixed logs—e.g., HCOPE/HCPI and SPIBB (Thomas et al., 2015; Laroche et al., 2019)—which bound deployment risk yet do not address how *expressive generators* should receive lower-tail gradients.

**Mixture-policy CVaR optimization.**  Luo et al. (Luo et al., 2024) propose a mixture policy parameterization for CVaR optimization in online RL, where a risk-neutral policy and an adjustable component are combined to form a risk-averse policy and improve the sample efficiency of CVaR policy gradients. While both their work and ours aim to shape the lower tail of returns, their method uses standard parametric actors and does not address offline data, generative policies, or explicit constraints on out-of-distribution actions. By contrast, RAMAC targets offline risk-sensitive control with a single expressive generative actor (diffusion/flow), a distributional critic, BC+CVaR objectives, per-state OOD constraints, and hazard relabeling.

**Closest lines and delineation** Concurrent actor–critic lines that couple diffusion with value learning remain expectation-oriented: (Zhang et al., 2025) stabilizes *online* diffusion actors with distributional critics and double-$Q$ but does not backpropagate CVaR along the denoising path ;(Fang et al., 2025)formulates *offline* constrained policy iteration as diffusion noise regression under KL/BC regularization ; and (Ma et al., 2025) studies efficient *online* diffusion control from an energy-based perspective. Distributional SAC variant (Ma et al., 2020) improve risk sensitivity via value-law estimation—typically with Gaussian policies—yet still lack CVaR shaping ; the diffusion-policy instantiation (Liu et al., 2025) targets multi-modality but likewise reports no CVaR along the multi-step generation . Risk-averse offline methods relying on behavior priors—e.g.(Urpí et al., 2021) ,and diffusion-prior (Chen et al., 2024b)—use anchor–perturb/mixing mechanisms , while (Ma et al., 2021) imposes conservative distributional critics (value pessimism) . These approaches either (i) optimize expectation-oriented objectives with expressive generators or (ii) control risk via mixing/pessimism, in contrast to our distributional risk shaping without anchor mixing.

## D  IMPLEMENTATION DETAILS

**Actor architecture** RAMAC employs a reparameterized generative actor $a = \psi_\theta(s, z)$ so that gradients from the risk term flow through the entire generative trajectory. RADAC instantiates $\psi_\theta$ as a denoising diffusion policy with VP schedule and $T{=}5$ denoising steps; the score network is an MLP (hidden 256–256, SiLU). RAFMAC instantiates $\psi_\theta$ as a deterministic flow–matching ODE solved by Euler with *flow_steps* $K{=}10$; the velocity field is an MLP (hidden 256–256, SiLU). For both, the actor objective is $\mathcal{L}_\pi = \lambda_{\mathrm{BC}}\mathcal{L}_{\mathrm{BC}} + \eta\,\mathcal{L}_{\mathrm{Risk}}$, where $\mathcal{L}_{\mathrm{BC}}$ is the model's native BC loss (score matching for diffusion, velocity matching for flow), and $\mathcal{L}_{\mathrm{Risk}} = -\mathbb{E}_{s,a\sim\pi_\theta}[\mathrm{CVaR}_\alpha(Z_\phi(s,a))]$ with $\alpha{=}0.1$.

**Distributional critic architecture** Both variants share a Double IQN critic trained with the quantile Huber loss ($\kappa{=}1$). Two critics $Z_{\phi_1}, Z_{\phi_2}$ are updated against a min target to curb overestimation; quantiles for TD use $\tau, \tau' \sim \mathcal{U}(0, 1)$, while the actor's CVaR term samples $\tau \sim U(0, \alpha)$.

For a batch $(s, a, r, s') \sim \mathcal{D}$, we generate the bootstrapping action via the actor:

$$a' = \psi_\theta(s', z'), \qquad z' \sim \mathcal{N}(0, I),$$

so that gradients (later used for risk shaping) can flow through the full generative trajectory (short reverse diffusion for RADAC; short ODE flow for RAFMAC). Right after specifying RADAC/RAFMAC actor parameters, we clarify the *pre-loss stage* for the critic before introducing the final loss. Instead of sampling $\tau$, we use a fixed uniform grid

$$\mathcal{T}_N = \left\{ \tau_i = \frac{i-\frac{1}{2}}{N} \right\}_{i=1}^N. \tag{16}$$

For CVaR at level $\alpha$, let $m = \lfloor \alpha N \rfloor$; then

$$\mathrm{CVaR}_\alpha\big(Z_\phi(s,a)\big) \approx \frac{1}{m}\sum_{i=1}^m Z_\phi(s,a;\tau_i), \qquad \tau_i \in \mathcal{T}_N. \tag{17}$$

This is equivalent in expectation to drawing $\tau \sim \mathcal{U}(0, \alpha)$ (cf. Eq. 9) but with lower estimator variance.

We form target quantiles on another grid $\mathcal{T}_{N'}$ and define the TD residual

$$\delta_{\tau_i, \tau'_j} = r + \gamma Z_{\bar{\phi}}(s', a'; \tau'_j) - Z_\phi(s, a; \tau_i), \qquad \tau_i \in \mathcal{T}_N,\ \tau'_j \in \mathcal{T}_{N'}.$$

With this pre-loss construction, the final critic objective is exactly the quantile-Huber residual minimization in Eq. 7/19; determinism only replaces stochastic $(\tau, \tau')$ by $(\tau_i, \tau'_j)$ from fixed grids.

The critic minimises the quantile-Huber loss (Dabney et al., 2018; Rowland et al., 2019)

$$\mathcal{L}_\kappa(\delta; \tau) = \big|\tau - \mathbf{1}_{\{\delta<0\}}\big| \times \begin{cases} \frac{\delta^2}{2\kappa}, & |\delta| \leq \kappa, \\ |\delta| - \frac{\kappa}{2}, & \text{otherwise,} \end{cases} \tag{18}$$

with $\kappa{=}1$. Averaging over $N \times N'$ quantile pairs yields

$$\mathcal{L}_{\mathrm{critic}}(\phi) = \mathbb{E}_{(s,a,r,s'),\,a'}\left[ \frac{1}{NN'}\sum_{i=1}^N\sum_{j=1}^{N'} \mathcal{L}_\kappa\big(\delta_{\tau_i,\tau'_j}; \tau_i\big) \right]. \tag{19}$$

Optimising Eq. 19 yields a calibrated estimate of the return law, whose lower tail supplies the CVaR gradients used in Step 2 (Sec. 3.2).

**Hyperparameters**    Unless noted, we use Adam for all networks (default $3 \times 10^{-4}$), batch size 256, discount $\gamma = 0.99$, soft target update $\tau_{\text{target}} = 0.005$, and no LR decay. RAMAC's (critic LR, IQN size, $\eta$, gradient–norm clipping, optional $Q$–target clipping, etc.) are listed in Table 7.

**RAFMAC risk weight tuning**    we swept $\eta \in \{1, 10, 50, 100, 300, 1000\}$ and *unified to* $\eta = 1000$ for all datasets; critic settings are fixed ($lr_{\text{critic}} = 3 \times 10^{-4}$, `emb_dim` = 128, `n_quantiles` = 32) (Table 7).

**Critic–target clipping**    Where specified , target returns are clipped ($[-300, 300]$ or $[-150, 150]$) to dampen rare outliers without affecting on–manifold learning.

# E    ADDITIONAL EXPERIMENTAL RESULTS

## E.1    MORE 2D SYNTHETIC TASK RESULTS

**Behavior cloning task Fig. 5**    On the 2D bandit dataset, three BC models show generator-specific patterns. CVAE-BC collapses topology and places probability in the low-density gap. Diffusion-BC most faithfully reproduces both the ring and the inner cluster with appropriate thickness. Flow-Matching BC draws a sharp ring but allocates less mass to the center and shows edges spread slightly outward. These baselines confirm that a suitably trained generative model can represent the full multimodal support of the dataset.

**RADAC dynamics over training Fig. 6**    RADAC starts from the same diffusion generator and adds the CVaR-based policy update on top of the BC objective. When the CVaR weight is set to zero (and RL guidance is disabled), RADAC reduces exactly to Diffusion-BC, recovering the same multimodal distribution as in Fig. 5. Thus, RADAC does *not* lack expressive power: the diffusion actor can represent both the high-return ring and the safe central cluster. Fig. 6 visualizes how the CVaR term reshapes this expressive policy over training. Early iterations spread mass over both modes; by ~200 epochs the policy begins to vacate the ring. Between 400 and 800 epochs the ring thins and probability shifts inward while the central cluster grows; by roughly 950 epochs most mass is at the safe center, matching the final snapshot in Fig. 3. The key point is that RADAC first *captures* the multimodal behavior distribution under BC, and only then *reallocates* mass toward the safest high-CVaR mode as low-quantile returns on the ring are penalized. The visually "almost unimodal" final policy is therefore a consequence of risk-aware optimization, not a limitation of the diffusion policy class.

**On conservatism and hyperparameters in prior baselines.**    The same Risky–Bandit geometry also helps explain why conservative tuning of prior methods (DiffusionQL, FlowQL, ORAAC-style anchor-perturbation) does not eliminate their structural failure modes. As defined in App. F.1, behavior actions lie on a thin, high-mean but heavy-tailed outer ring and a lower-mean but light-tailed central Gaussian cluster. Geometrically, each behavior anchor $b$ lies on a narrow manifold with margin $m(b)$ to the low-density gap and to regions that are unobserved in the dataset. Once a perturbation ball $\mathbb{B}_\varepsilon(b)$ has $\varepsilon \gtrsim m(b)$, it necessarily crosses into these off-support regions. In this regime, residual or anchor-perturbation policies are not constrained to stay tangent to the data manifold, so even small residuals can systematically produce OOD actions, which in turn leads to unstable Q-estimates and higher exposure to the trap-heavy ring. This is precisely the behavior illustrated in Fig. 3(f–g).

For DiffusionQL and FlowQL, the RL term is always driven by a risk-neutral Q-function. Sweeping the RL scaling coefficient $\eta$ (as shown in Fig. 7) shows the expected dichotomy: as $\eta \to 0$ the policies revert to BC-like behavior, recovering the multimodal density of the behavior data; as $\eta$ increases, mass increasingly concentrates on the high-mean ring, despite its heavy lower tail. FlowQL's flow-matching prior retains a bit more mass near the center, but the dominant mode still

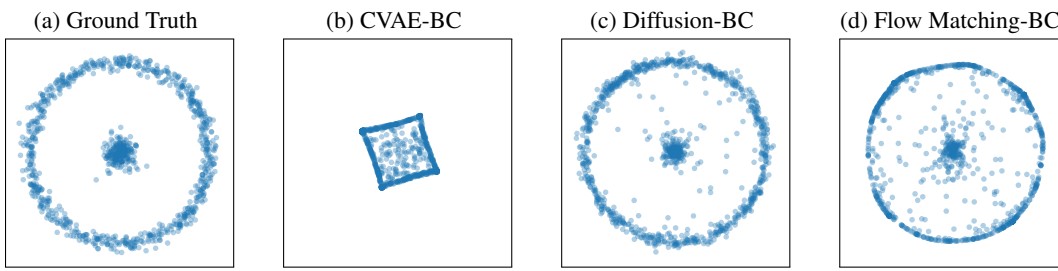

Figure 5: **Behavior cloning on the Risk Bandit dataset.** Each panel shows i.i.d. samples from the BC Policy. CVAE-BC mixes modes and places points in the low-density gap; Diffusion-BC reproduces both the outer ring and the central cluster; Flow-Matching BC yields a crisp ring but assigns less mass to the center.

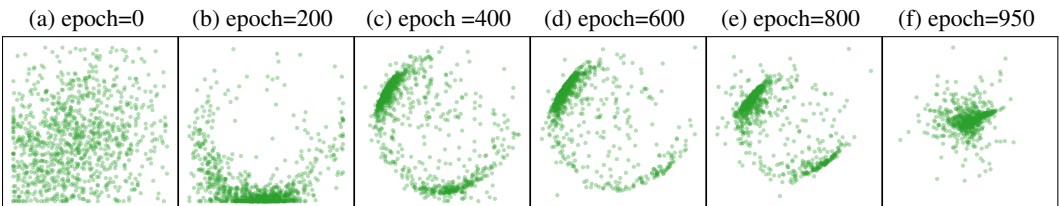

Figure 6: **RADAC dynamics on the toy task.** Mass gradually moves from the risky ring to the safe center: the ring thins (400–800 epochs) and the central cluster grows, ending with most mass at the center ( 950 epochs). BC keeps the policy on-manifold while CVaR reduces lower-tail risk.

tracks the risky ring. No choice of $\eta$ can simultaneously prevent this tendency and retain a nontrivial RL component, because the objective is fundamentally risk-neutral.

Fig. 8 shows ORAAC-style anchor-perturbation exhibits an analogous tradeoff. When the risk-distortion weight $\lambda$ is set to zero, the method again collapses to BC on the behavior manifold. As $\lambda$ increases, the residual pushes probability mass away from the BC solution: in the Risky–Bandit geometry this either splits modes or routes probability through low-density regions between them, and for larger $\lambda$ the policy can "jump" toward arbitrary corners of the action space, far from any behavior support (as predicted by the analysis in Sec. 4).

In other words, the observed failures of Fig. 3(c-g) are not artifacts of insufficient hyperparameter tuning but a structural consequence of combining risk-neutral or anchor-perturbation objectives with thin, non-convex data supports. RADAC replaces this with an on-manifold, risk-aware update that reduces tail risk without relying on off-support residuals.

For completeness, we also sweep the RL weight $\eta$ in RADAC on the same Risky–Bandit (Fig. 9). When $\eta \to 0$, the objective reduces to pure BC and the diffusion policy faithfully matches the multi-modal behavior distribution, reproducing both the outer ring and the central cluster. As $\eta$ increases, the CVaR-guided term gradually reallocates probability mass away from the high-mean but heavy-tailed ring toward the light-tailed central mode while staying on-support: the ring thins and eventually disappears, leaving a compact cluster around the safe center. Unlike DiffusionQL/FlowQL and ORAAC-style anchor–perturbation, this sweep does not create spurious off-support modes or low-density bridges; instead, it yields a smooth trade-off between ring coverage and tail-risk reduction, consistent with the on-manifold, risk-aware behavior-regularized objective in Eq. 12.

### E.2    EXTENDED STOCHASTIC-D4RL RESULTS

**Protocol**    To remove post-hoc checkpoint selection and ease reproducibility, we report a full result in Sec. 3.1 with s.e. in Table 3 and *fixed 1000-gradient-step evaluation* for every method and task in Table 4. Scores are raw returns and episodic $\text{CVaR}_{0.1}$ (mean $\pm$ s.e. over 5 seeds), without normalization, matching the stochastic variants used in the main text.

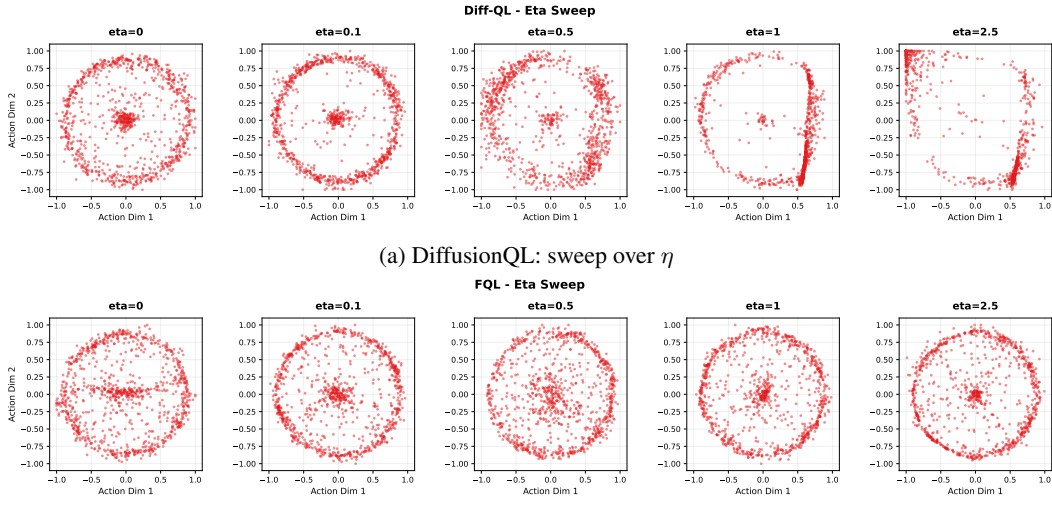

(a) DiffusionQL: sweep over $\eta$

(b) FlowQL: sweep over $\eta$

Figure 7: **Effect of RL weight $\eta$ in Risky-Bandit.** Each panel shows a policy trained on the 2D Risky–Bandit with a different RL scaling coefficient $\eta$. As $\eta \to 0$, especially DiffusionQL reverts to BC-like behavior and recover the multimodal behavior density. As $\eta$ increases, probability mass concentrates on the high-mean outer ring despite its heavy lower tail, illustrating the risk-neutral tendency discussed in Sec. 3 and App. E.

**Consistency with the main-text trends**   At the fixed 1000–step evaluation, the ranking patterns largely match the main text, but the mechanisms are task–dependent. Flow–based policies (FlowQL/RAFMAC) often reach higher mean by 1000 steps because flow matching uses a deterministic ODE with a short generative path and low-variance policy gradients; combined with velocity-matching BC, this yields fast on-manifold improvement. CVaR outcomes depend on lower-tail calibration of the distributional critic: with smooth, non-terminating penalties (e.g., HalfCheetah) RADAC/RAFMAC already improve CVaR at 1000 steps, whereas with sparse, terminating hazards (e.g., Hopper) ORAAC's anchor regularization provides more stable early CVaR and mean. Walker2d sits in between: RAFMAC attains the highest mean at 1000 steps, and CVaR leadership alternates between FlowQL and RAFMAC depending on the dataset variant.

**Pareto Frontier Analysis: Return vs. Safety Violations**   Figure 10 plots mean return (y) against safety-violation counts per episode (x), with color indicating training progress. Unless noted, comparisons refer to the same 1000-step evaluation as in Table 4. We organize the discussion by algorithm.

Across datasets, RADAC populates the upper-left region of the frontier: for comparable return, it tends to incur fewer violations. Only for HALFCHEETAH–medium–expert, RADAC sometimes drifts up-right (higher return with slightly more violations) because the penalty is light and non-terminating, so near-threshold speed pays off, consistent with its best Mean/CVaR. Mechanistically, diffusion with CVaR guidance enables fine-grained reweighting away from safety thresholds while BC keeps samples on-manifold, so trajectories in the plot drift left (fewer violations) without sacrificing return. RAFMAC pushes the top of the frontier in mean—most clearly on WALKER2D and HALFCHEETAH-m-r—and is competitive in CVaR (Table 4). Deterministic ODE transport with low-variance policy gradients and velocity-matching BC yields fast on-manifold improvement; in the Pareto view this appears as high-return points with modest violation counts. Because the transport is geometry-preserving, boundary-adjacent mass tends to thin rather than disappear abruptly; CVaR improves as the velocity field adapts. ORAAC forms the frontier on HOPPER-m-e with few violations and strong returns, matching its leading scores under terminating pose hazards. In other settings it remains reliably conservative (low violations) at the cost of mean on some tasks, consistent with anchor-based regularization. FlowQL often achieves high-mean points but with comparatively higher violation counts in the Pareto plot. Without tail-aware guidance, safety depends on the expected-value critic and task smoothness, explaining the variability across datasets. DiffusionQL

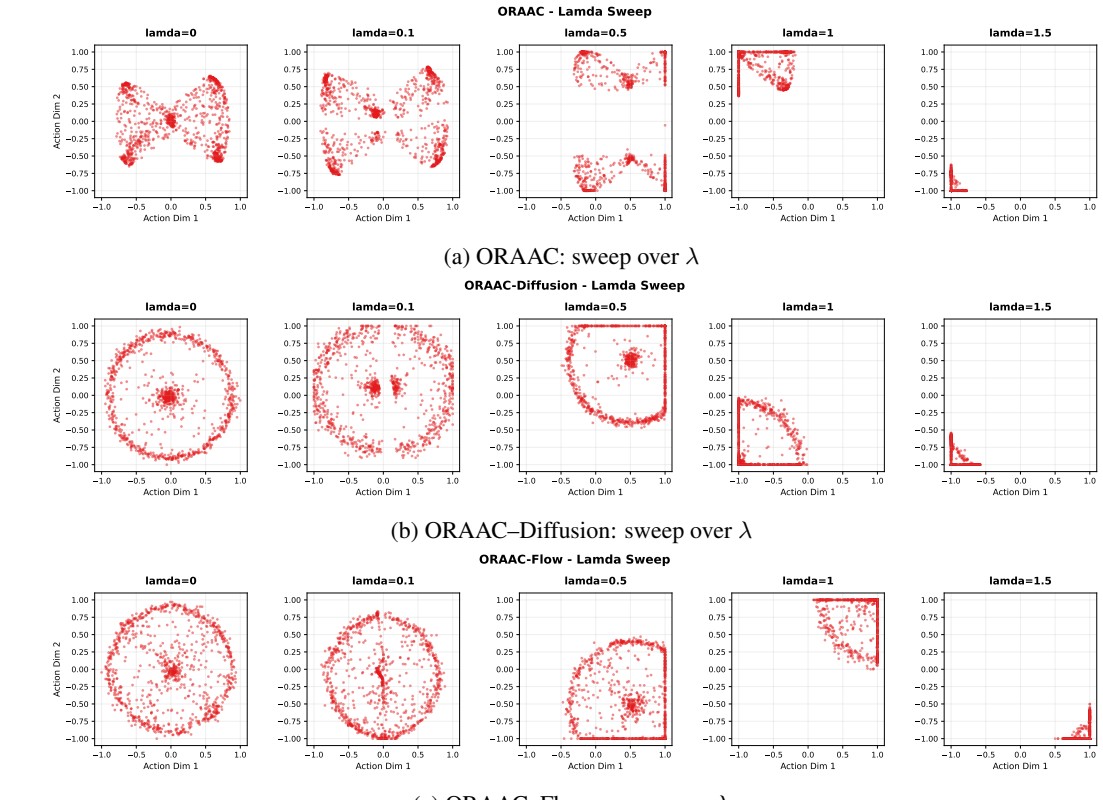

(a) ORAAC: sweep over $\lambda$

(b) ORAAC–Diffusion: sweep over $\lambda$

(c) ORAAC–Flow: sweep over $\lambda$

Figure 8: **Effect of risk weight $\lambda$ in anchor-perturbation baselines.** Each panel visualizes policies on the 2D Risky–Bandit as the risk-distortion weight $\lambda$ is swept. For $\lambda = 0$, all ORAAC variants reduce to BC on the behavior manifold. As $\lambda$ increases, residual updates push probability mass away from the BC solution: modes can split or leak through low-density gaps, and for larger $\lambda$ the policy may jump toward regions far from any behavior support, consistent with the analysis in Sec. 4.

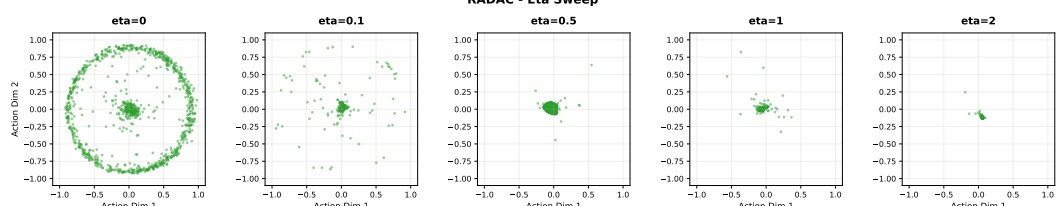

Figure 9: **Effect of RL weight $\eta$ in RADAC.** As $\eta$ increases from 0 (pure BC), probability mass smoothly moves from the risky outer ring to the safe central cluster, without creating off-support modes.

exhibits wider scatter: runs either reach moderate returns with elevated violations or collapse to low-return, near-zero violation regions. This variability is consistent with value-only guidance under stochastic penalties and matches its weaker CVaR. CODAC clusters in the low-return/low-violation corner across tasks, as expected from conservative critics.

### E.3 ABLATION STUDY

We evaluate RADAC and RAFMAC with three risk distortions CVaR, Wang, and CPW under the same 1000-step evaluation protocol used above. Across seeds, Wang generally tilts updates toward

Table 3: Stochastic D4RL (200/500step evaluation): (Mean and CVaR$_{0.1} \pm$ s.e. over 5 seeds).

| Environment, Dataset | Algorithm | Mean | CVaR |
|---|---|---|---|
| HalfCheetah-m-e | CQL | $-66.66 \pm 13.17$ | $-135.39 \pm 27.71$ |
| | CODAC | $-0.12 \pm 0.16$ | $-0.11 \pm 0.25$ |
| | ORAAC | $796.06 \pm 30.28$ | $742.94 \pm 22.95$ |
| | FlowQL | $844.14 \pm 16.15$ | $754.44 \pm 27.26$ |
| | DiffusionQL | $-20.71 \pm 18.89$ | $-76.39 \pm 14.39$ |
| | RAFMAC | $889.56 \pm 38.31$ | $736.95 \pm 102.54$ |
| | RADAC | $\mathbf{916.64} \pm 35.80$ | $\mathbf{805.25} \pm 15.34$ |
| Walker2d-m-e | CQL | $-21.52 \pm 8.68$ | $-64.88 \pm 18.32$ |
| | CODAC | $23.96 \pm 10.56$ | $-43.88 \pm 13.28$ |
| | ORAAC | $969.62 \pm 442.36$ | $358.55 \pm 682.29$ |
| | FlowQL | $1309.48 \pm 233.72$ | $468.15 \pm 416.61$ |
| | DiffusionQL | $-32.38 \pm 64.15$ | $-116.19 \pm 46.39$ |
| | RAFMAC | $\mathbf{1822.24} \pm 128.36$ | $\mathbf{1127.21} \pm 620.95$ |
| | RADAC | $1708.68 \pm 163.19$ | $573.22 \pm 894.62$ |
| Hopper-m-e | CQL | $-25.87 \pm 13.46$ | $-111.37 \pm 51.69$ |
| | CODAC | $26.59 \pm 47.56$ | $-150.92 \pm 42.19$ |
| | ORAAC | $\mathbf{714.15} \pm 243.57$ | $\mathbf{374.63} \pm 326.66$ |
| | FlowQL | $341.16 \pm 75.98$ | $-8.80 \pm 84.57$ |
| | DiffusionQL | $-279.97 \pm 215.46$ | $-872.95 \pm 589.90$ |
| | RAFMAC | $281.24 \pm 82.07$ | $-132.33 \pm 183.92$ |
| | RADAC | $130.74 \pm 273.53$ | $-167.29 \pm 107.33$ |
| HalfCheetah-m-r | CQL | $-66.21 \pm 11.52$ | $-127.09 \pm 37.10$ |
| | CODAC | $-0.11 \pm 0.16$ | $-1.47 \pm 0.53$ |
| | ORAAC | $18.99 \pm 34.67$ | $-34.09 \pm 25.47$ |
| | FlowQL | $434.33 \pm 40.45$ | $224.73 \pm 146.83$ |
| | DiffusionQL | $279.95 \pm 91.48$ | $79.93 \pm 110.85$ |
| | RAFMAC | $449.04 \pm 73.84$ | $144.73 \pm 181.54$ |
| | RADAC | $\mathbf{525.84} \pm 44.61$ | $\mathbf{278.65} \pm 151.27$ |
| Walker2d-m-r | CQL | $-16.90 \pm 7.56$ | $-51.49 \pm 14.17$ |
| | CODAC | $33.59 \pm 45.29$ | $-52.63 \pm 42.63$ |
| | ORAAC | $126.94 \pm 178.91$ | $-203.64 \pm 338.87$ |
| | FlowQL | $411.36 \pm 70.84$ | $5.08 \pm 240.85$ |
| | DiffusionQL | $96.88 \pm 198.31$ | $48.14 \pm 227.71$ |
| | RAFMAC | $-71.69 \pm 241.69$ | $\mathbf{530.37} \pm 84.57$ |
| | RADAC | $\mathbf{615.94} \pm 219.44$ | $145.21 \pm 39.43$ |
| Hopper-m-r | CQL | $-16.25 \pm 20.60$ | $-118.70 \pm 106.89$ |
| | CODAC | $-47.83 \pm 32.01$ | $-160.08 \pm 60.90$ |
| | ORAAC | $-18.00 \pm 44.92$ | $-129.25 \pm 108.63$ |
| | FlowQL | $373.16 \pm 109.86$ | $-62.24 \pm 203.02$ |
| | DiffusionQL | $-2.79 \pm 12.83$ | $-51.33 \pm 36.90$ |
| | RAFMAC | $303.44 \pm 28.95$ | $-90.73 \pm 93.82$ |
| | RADAC | $\mathbf{385.58} \pm 55.20$ | $\mathbf{-8.16} \pm 92.79$ |

higher means and weaker tails; CPW sits between CVaR and Wang but shows higher variance across seeds. Overall, CVaR is the most reliable choice for lower-tail control at comparable mean.

### E.4 OOD DETECTOR ROBUSTNESS

To check that the RADAC < ORAAC trend in Table 2 is not an artifact of the 1-NN score, we repeated the analysis with two additional detectors: a Local Outlier Factor (LOF) and a simple Mahalanobis detector based on a single Gaussian fit to the dataset actions. Table 6 reports the three Stochastic-D4RL medium-expert tasks with these detectors. On HALFCHEETAH and WALKER2D, all three detectors agree with the main-text result and rank RADAC as having substantially lower OOD action rates than ORAAC. On HOPPER, RADAC still achieves $5$–$8\times$ lower OOD rates under 1-NN and LOF, while we see one exception: the single-Gaussian Mahalanobis detector returns $0\%$ OOD for ORAAC. This reflects a limitation of the detector rather than a reversal of the trend. ORAAC on this task collapses most of its mass onto a single expert-like mode with very small variance, so that all of its actions lie deep inside the $99\%$ confidence ellipsoid of a Gaussian fit to the dataset. A global Mahalanobis score cannot see the thin or multi-modal structure of the dataset manifold and therefore misses the local leakage that is still picked up by the 1-NN and LOF detectors, for which ORAAC continues to have higher OOD rates than RADAC. Aggregating over three seeds, RADAC retains lower 1-NN and LOF OOD rates than ORAAC on all three tasks, and lower Mahalanobis

Table 4: Stochastic D4RL (1000-step evaluation): Mean and $\text{CVaR}_{0.1}\pm$ s.e. over 5 seeds.

| Environment, Dataset | Algorithm | Mean | CVaR |
|---|---|---|---|
| HalfCheetah-m-e | CQL | $-0.97\pm0.24$ | $-2.24\pm0.43$ |
| | CODAC | $-0.12\pm0.08$ | $-1.48\pm0.27$ |
| | ORAAC | $4106.25\pm177.48$ | $3692.79\pm466.31$ |
| | FlowQL | $4695.46\pm65.97$ | $4025.12\pm230.08$ |
| | DiffusionQL | $-118.72\pm64.53$ | $-198.01\pm76.76$ |
| | RAFMAC | $5084.12\pm230.43$ | $3735.37\pm827.60$ |
| | RADAC | $5659.40\pm131.94$ | $4667.96\pm42.59$ |
| Walker2d-m-e | CQL | $-10.32\pm6.27$ | $-73.38\pm9.02$ |
| | CODAC | $27.56\pm6.26$ | $-35.30\pm15.36$ |
| | ORAAC | $663.23\pm181.31$ | $205.21\pm65.45$ |
| | FlowQL | $2457.68\pm208.80$ | $448.48\pm208.81$ |
| | DiffusionQL | $-32.33\pm4.59$ | $-68.43\pm11.28$ |
| | RAFMAC | $3567.89\pm206.63$ | $356.20\pm987.34$ |
| | RADAC | $2760.21\pm689.32$ | $322.76\pm757.44$ |
| Hopper-m-e | CQL | $43.22\pm29.48$ | $-65.90\pm36.42$ |
| | CODAC | $31.59\pm28.74$ | $-77.88\pm34.33$ |
| | ORAAC | $660.07\pm157.55$ | $400.84\pm142.60$ |
| | FlowQL | $393.64\pm27.75$ | $77.93\pm60.53$ |
| | DiffusionQL | $-38.75\pm27.68$ | $-212.49\pm91.99$ |
| | RAFMAC | $370.11\pm39.95$ | $-120.09\pm56.34$ |
| | RADAC | $-764.93\pm741.86$ | $-1094.93\pm806.85$ |
| HalfCheetah-m-r | CQL | $-38.85\pm38.44$ | $-40.23\pm38.44$ |
| | CODAC | $-0.12\pm0.08$ | $-1.48\pm0.26$ |
| | ORAAC | $315.87\pm69.27$ | $161.54\pm68.76$ |
| | FlowQL | $1909.57\pm395.55$ | $568.43\pm256.85$ |
| | DiffusionQL | $2261.16\pm531.18$ | $1439.77\pm461.28$ |
| | RAFMAC | $2696.61\pm110.68$ | $1499.80\pm394.08$ |
| | RADAC | $2674.72\pm51.76$ | $1401.03\pm199.08$ |
| Walker2d-m-r | CQL | $-14.68\pm5.52$ | $-95.30\pm18.50$ |
| | CODAC | $26.39\pm7.97$ | $-36.56\pm12.92$ |
| | ORAAC | $160.23\pm147.55$ | $-359.49\pm302.72$ |
| | FlowQL | $647.33\pm166.12$ | $-29.64\pm110.73$ |
| | DiffusionQL | $-23.50\pm4.44$ | $-53.55\pm12.30$ |
| | RAFMAC | $778.00\pm130.03$ | $7.92\pm35.77$ |
| | RADAC | $383.87\pm288.95$ | $-309.70\pm246.62$ |
| Hopper-m-r | CQL | $2.28\pm42.17$ | $-130.48\pm53.25$ |
| | CODAC | $3.61\pm18.41$ | $-105.41\pm19.86$ |
| | ORAAC | $-30.00\pm32.77$ | $-179.92\pm61.46$ |
| | FlowQL | $448.26\pm70.39$ | $-33.21\pm43.38$ |
| | DiffusionQL | $-22.15\pm24.93$ | $-163.82\pm59.18$ |
| | RAFMAC | $350.36\pm33.05$ | $-36.69\pm28.35$ |
| | RADAC | $453.64\pm68.46$ | $-87.04\pm123.96$ |

Table 5: Ablation (1000-step evaluation). RADAC/RAFMAC with CVaR, Wang, and CPW on HALFCHEETAH-medium-replay and WALKER2D-medium-replay. Scores are mean $\pm$ s.e. over 3 seeds.

| Method | Distortion | HalfCheetah–m–r | | Walker2d–m–r | |
|---|---|---|---|---|---|
| | | Mean | $\text{CVaR}_{0.1}$ | Mean | $\text{CVaR}_{0.1}$ |
| RADAC | CVaR | $2758.5 \pm 84.1$ | $1759.5 \pm 71.5$ | $681.3 \pm 409.3$ | $-395.1 \pm 438.3$ |
| RADAC | Wang | $2653.5 \pm 86.5$ | $310.8 \pm 92.6$ | $417.3 \pm 397.0$ | $-52.1 \pm 11.4$ |
| RADAC | CPW | $2777.9 \pm 93.7$ | $1061.6 \pm 731.7$ | $64.3 \pm 149.3$ | $-203.6 \pm 69.8$ |
| RAFMAC | CVaR | $2835.8 \pm 116.3$ | $1981.2 \pm 405.3$ | $698.8 \pm 215.5$ | $5.6 \pm 60.8$ |
| RAFMAC | Wang | $2625.6 \pm 113.8$ | $462.5 \pm 427.6$ | $552.2 \pm 134.8$ | $-706.4 \pm 687.5$ |
| RAFMAC | CPW | $2539.2 \pm 31.1$ | $95.9 \pm 92.3$ | $360.7 \pm 49.6$ | $-71.6 \pm 22.1$ |

OOD rates on two out of three tasks, indicating that the RADAC < ORAAC ordering is robust to detector. Implementation details of these detectors are detailed in App. F.4

### E.5 RUNTIME AND INFERENCE LATENCY

We compare wall-clock training time and per-action inference latency for DiffusionQL, FlowQL, and RADAC on HOPPER-MEDIUM-EXPERT-V2. All methods use the same A6000 GPU , PyTorch/CUDA stack, and training budget. Training time is measured between the first and last gra-

Figure 10: Pareto frontiers of return vs. safety violations. Rows are Stochastic–D4RL tasks (top→bottom: HALFCHEETAH-m-e, HALFCHEETAH-m-r, WALKER2D-m-e, WALKER2D-m-r, HOP-PER-m-e, HOPPER-m-r); columns are algorithms (left→right: RADAC, DiffusionQL, FlowQL, ORAAC, CODAC). Points are evaluation snapshots across training (color encodes epoch; dark→yellow). $x$–axis: violation count per episode; $y$–axis: mean return (upper–left is better)

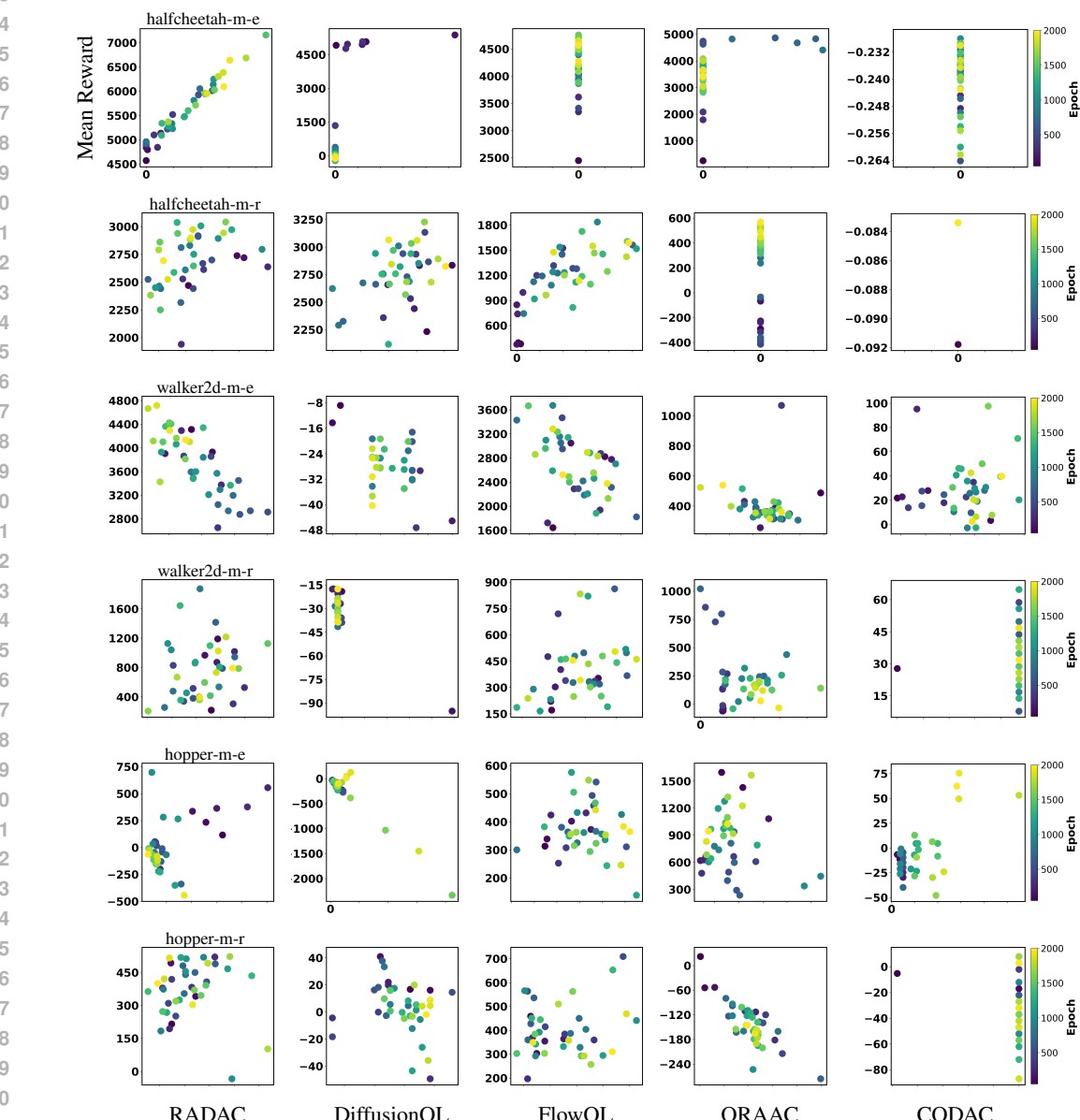

dient update; latency is the average GPU time to produce a single action for $10^4$ replay states with `torch.no_grad`.

RADAC replaces the scalar critic with a distributional IQN and optimizes a CVaR objective, but this does not dominate runtime. In these methods, most of the computational cost comes from the expressive generative actor (diffusion or flow), not from the critic. The extra work required by IQN, evaluating a small number of quantiles and averaging the bottom $\alpha$-fraction for CVaR, adds only minor overhead relative to a full diffusion / flow pass.

Table 6: Detector–robust OOD action rates $\varepsilon_{\text{act}}$ (%, mean $\pm$ s.e. over 3 seeds) on Stochastic–D4RL medium–expert tasks. Smaller is better.

| Env | Algo | 1-NN | LOF | Mahalanobis |
|---|---|---|---|---|
| HalfCheetah-m.e. | RADAC | $2.01 \pm 0.31$ | $4.42 \pm 0.94$ | $0.09 \pm 0.02$ |
| HalfCheetah-m.e. | ORAAC | $65.24 \pm 4.54$ | $10.02 \pm 1.72$ | $0.34 \pm 0.33$ |
| Hopper-m.e. | RADAC | $0.68 \pm 0.06$ | $2.09 \pm 0.21$ | $0.43 \pm 0.03$ |
| Hopper-m.e. | ORAAC | $3.71 \pm 2.52$ | $14.23 \pm 1.45$ | $0.00 \pm 0.00$ |
| Walker2d-m.e. | RADAC | $1.47 \pm 0.42$ | $0.74 \pm 0.13$ | $0.50 \pm 0.12$ |
| Walker2d-m.e. | ORAAC | $2.80 \pm 0.50$ | $5.81 \pm 1.46$ | $0.76 \pm 0.26$ |

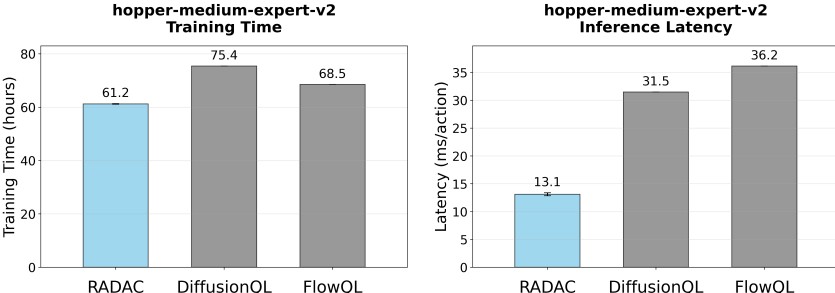

Figure 11: Wall-clock training time (left) and per-action inference latency (right) on HOPPER-MEDIUM-EXPERT-V2.

Conversely, FlowQL carries its own overhead by training both a flow-matching prior and a distilled one-step policy head, and DiffusionQL implementation uses slightly heavier hyperparameters (e.g., more denoising steps and a larger actor) than RADAC. As a result, under our implementation and hyperparameter choices, RADAC ends up slightly faster in wall-clock time on HOPPER-MEDIUM-EXPERT-V2. We do not claim that RADAC is intrinsically faster than DiffusionQL or FlowQL; these measurements simply show that the CVaR + distributional critic extension does *not* introduce an order-of-magnitude runtime penalty. Inference latency is likewise dominated by the shared diffusion/flow backbone, and the RADAC actor achieves comparable or lower per-action latency than the risk-neutral expressive baselines when using similar numbers of denoising / flow steps.

### E.6 EFFECT OF THE TAIL SAMPLE SIZE $K$ AND $N$ ON THE CVAR ESTIMATOR

To clarify the role of the tail sample size $K$ in our IQN-based CVaR estimator, we run an experiment on a subset of D4RL, HALFCHEETAH-MEDIUM-REPLAY-V2 and WALKER2D-MEDIUM-REPLAY-V2. During training we randomly sample 2,000 states from the replay buffer and store them as a fixed evaluation set $\{s_j\}_{j=1}^{2000}$. For each state $s_j$, each $K \in \{2, 4, 8, 16\}$, and each risk level $\alpha \in \{0.05, 0.1, 0.2\}$ we estimate $\text{CVaR}_\alpha$ using the offline-selected RADAC policy (trained with $\alpha_{\text{train}} = 0.1$).

Given a state $s$, we draw $K$ i.i.d. pairs $(a_k, \tau_k)$ with $a_k \sim \pi_\theta(\cdot \mid s)$ and $\tau_k \sim \text{Unif}(0, \alpha)$, evaluate both distributional critics $Z_{\phi_1}, Z_{\phi_2}$, and form the tail-sampling CVaR estimator

$$\widehat{\text{CVaR}}_\alpha(s) = \frac{1}{K} \sum_{k=1}^{K} \min_{i \in \{1,2\}} Z_{\phi_i}(s, a_k; \tau_k). \tag{20}$$

For each $(K, \alpha)$ we repeat this procedure 100 times per state (using new $(a_k, \tau_k)$ draws each time), compute the variance of $\widehat{\text{CVaR}}_\alpha(s_j)$ across the 100 repetitions for each state $s_j$, and then aggregate these per-state variances by their mean and standard deviation over $j = 1, \ldots, 2000$.

Figure 12 reports the resulting *estimator variance* as a function of $K$ for $\alpha \in \{0.05, 0.1, 0.2\}$. Across both tasks and all three risk levels the variance decreases approximately at a $1/K$ rate: using only $K = 2$ tail samples yields noisy CVaR estimates, while increasing $K$ to 4 and 8 substantially

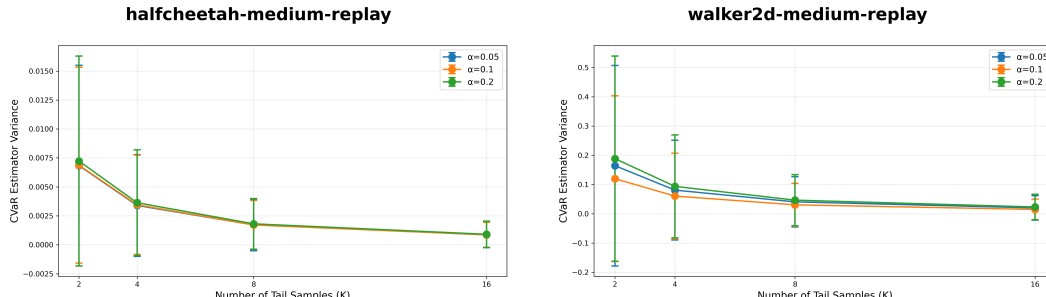

Figure 12: Empirical variance of the IQN-based $\mathrm{CVaR}_{0.1}$ estimator as a function of the tail sample size $K$ for the offline-selected RADAC policies on HALFCHEETAH-MEDIUM-REPLAY-V2 (left) and WALKER2D-MEDIUM-REPLAY-V2 (right). For each $K \in \{2, 4, 8, 16\}$ we compute $\widehat{\mathrm{CVaR}_\alpha}$; the curves show the mean per-state estimator variance with error bars indicating one standard deviation across states for $\alpha \in \{0.05, 0.1, 0.2\}$; all three risk levels exhibit a similar $1/K$-like decay.

reduces the variance. The marginal improvement from $K = 8$ to $K = 16$ is much smaller, despite doubling the number of tail samples. This supports our choice of a moderate tail size (with $K$ in the range 8–16 across tasks in our experiments), which provides a good trade-off between estimator noise and computational cost. Note that the absolute variance scale differs between environments.

**Role of the number of quantiles $N$.** The tail estimator in Eq. 20 depends on the number of tail samples $K$, while the IQN critic itself is trained with $N$ quantile samples per state–action pair. In all experiments we use moderate values $N \in \{16, 32\}$ (see Table. 7), which are standard in prior IQN-based work and make the critic updates stable. Increasing $N$ primarily reduces the variance of the critic update and smooths the learned value distribution, but once $N$ is in this moderate range we do not observe qualitative changes in the CVaR estimates or policy performance, while the computational cost grows roughly linearly in $N$. Hence we treat $N$ as a fixed architectural hyperparameter.

### E.7 RETURN DISTRIBUTIONS OF ROLLOUT TRAJECTORIES

We here visualize the empirical return distributions under the stochastic hazard wrapper for the main Stochastic-D4RL tasks. For each method and environment we aggregate 30–60 evaluation episodes across three seeds and plot histograms and kernel-density estimates of the rollout returns, together with vertical markers for the mean, median, and $\mathrm{CVaR}_{0.1}$ (Fig. 13). On HALFCHEETAH-MEDIUM-EXPERT-V2, for example, RADAC concentrates mass in a high-return band ($\mathrm{CVaR}_{0.1} \approx 4.7 \times 10^3$) while ORAAC exhibits a multi-modal distribution with occasional near-zero or negative episodes, and DiffusionQL/CQL collapse near zero. Since hazard events in our wrapper correspond to large negative returns and early termination, catastrophic hazard-inducing episodes populate the extreme left tail; the reduced tail mass for RADAC thus reflects a lower frequency of hazard-heavy trajectories rather than mere over-conservatism.

### E.8 RISK-RETURN FRONTIER UNDER CVaR LEVEL

To make clearer view of the safety/return trade-off, we visualize how the CVaR level $\alpha$ affects RADAC's behavior on two representative tasks: HALFCHEETAH-MEDIUM-REPLAY-V2 and WALKER2D-MEDIUM-REPLAY-V2. For each environment, we train RADAC with $\alpha \in \{0.05, 0.10, 0.20\}$ using the same hyperparameters as in the main Stochastic-D4RL experiments, and aggregate results across multiple seeds. For each run, we select the checkpoint with the highest normalized score and compute the mean normalized return and empirical $\mathrm{CVaR}_\alpha$ from eval rollouts under the risky wrapper. We then report the seed-averaged mean normalized score versus $\mathrm{CVaR}_\alpha$, with error bars denoting the standard error of the mean across seeds (Fig. 14).

On HALFCHEETAH-MEDIUM-REPLAY-V2, varying $\alpha$ across $\{0.05, 0.10, 0.20\}$ primarily affects the tail: the points move noticeably along the horizontal axis in $\mathrm{CVaR}_\alpha$, while the mean normalized score stays in a very tight band, indicating that RADAC can reshape the lower tail of the return dis-

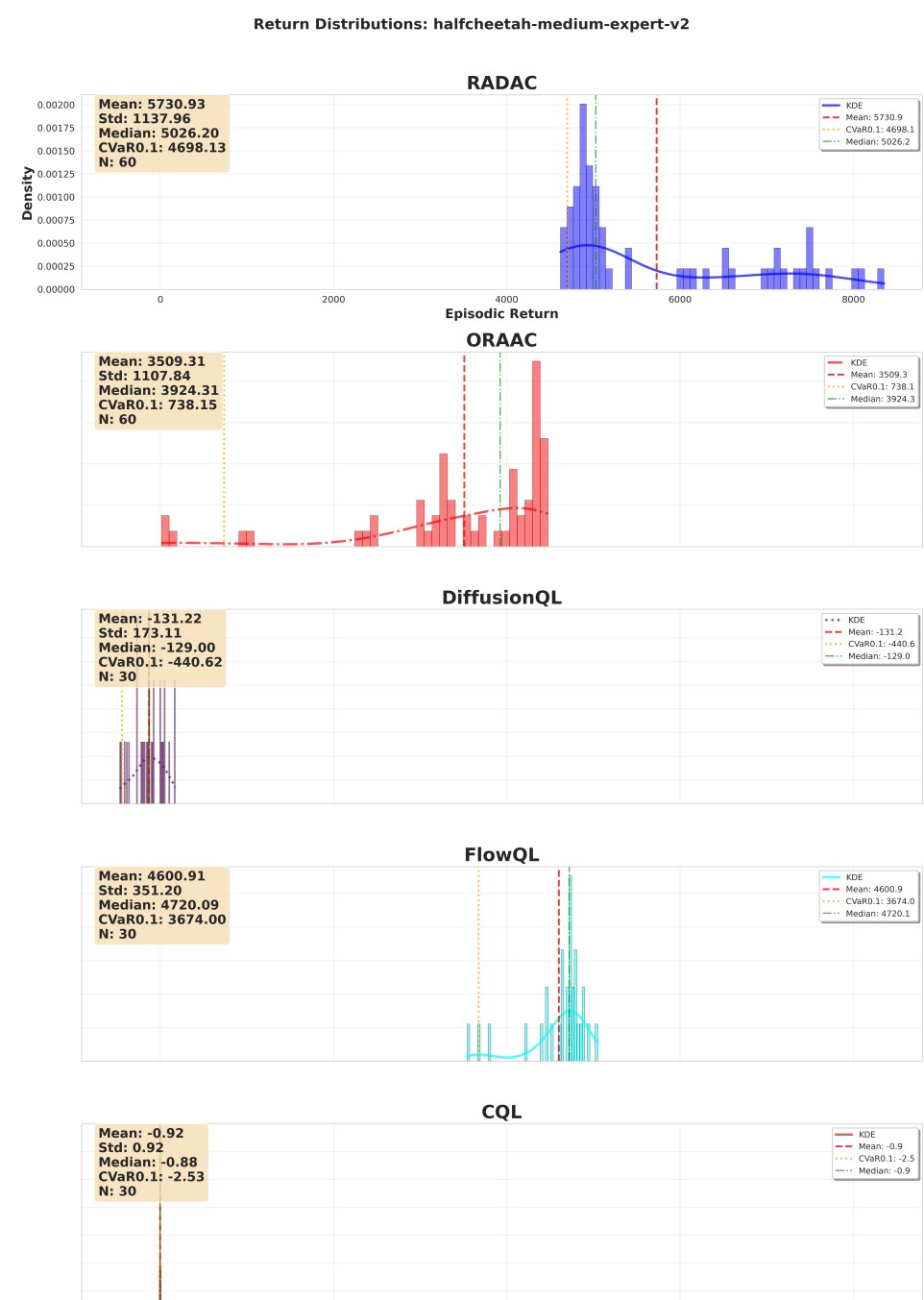

Figure 13: Empirical return distributions on HALFCHEETAH-MEDIUM-EXPERT-V2 under the stochastic hazard wrapper. Each subplot shows a histogram and KDE of episodic returns across evaluation rollouts, with vertical lines for the mean, median, and $\mathrm{CVaR}_{0.1}$.

tribution with only a minor impact on average performance. In contrast, on WALKER2D-MEDIUM-REPLAY-V2 the frontier moves up and to the right as we adjust $\alpha$: the more tail-sensitive settings simultaneously improve $\mathrm{CVaR}_\alpha$ and the mean normalized score, suggesting that on this task catastrophic low-return trajectories are frequent enough that suppressing them not only reduces tail risk but also raises average returns. Overall, these frontiers confirm that the BC+CVaR objective in

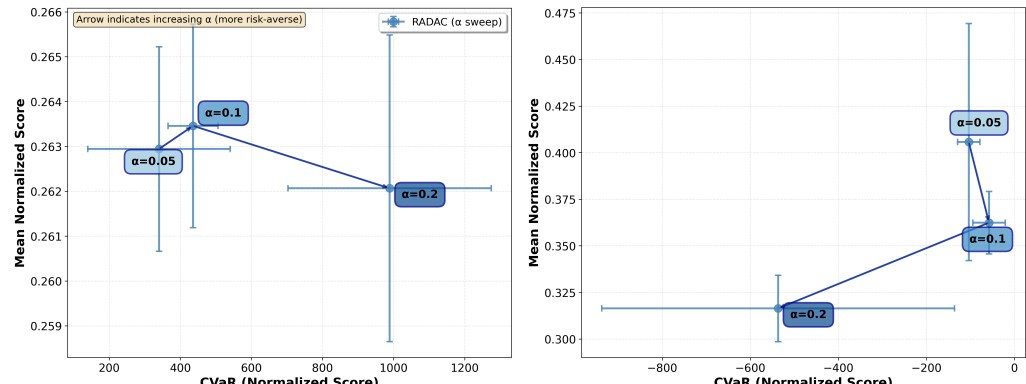

Figure 14: **Risk-return frontier for RADAC under different CVaR levels.** Each point shows the seed-averaged mean normalized score and $\mathrm{CVaR}_\alpha$ for RADAC trained with a fixed $\alpha \in \{0.05, 0.10, 0.20\}$ on HALFCHEETAH-MEDIUM-REPLAY-V2 (left) and WALKER2D-MEDIUM-REPLAY-V2 (right). Error bars denote the standard error of the mean across seeds.

RADAC exposes a smooth knob to trade off tail-risk and mean return, and that in some regimes increasing tail sensitivity can strictly improve both safety and performance.

## F EXPERIMENTAL DETAILS

### F.1 2D SYNTHETIC TASK DETAILS

**Risky–Bandit dataset** We generate $N = 10^4$ state–action–reward tuples with dummy zero states. Actions come from two modes: (i) Ring (80%): radius $0.9 \pm 0.04$; base reward $\mathcal{N}(9, 0.3^2)$; with probability 0.05 a trap penalty $-40$ is applied (heavy lower tail). (ii) Centre (20%): $\mathcal{N}(\mathbf{0}, 0.1^2\mathbf{I})$; reward $\mathcal{N}(5, 0.3^2)$. Actions are clipped to $[-1, 1]^2$.

All methods train on the same static dataset; when a BC regulariser is required we use the standard loss of the underlying generator. RADAC adds the CVaR term from Eq. 12 to the diffusion/flow BC objective and backpropagates. For each trained policy we draw 1,000 action samples for visualisation in Fig. 3.

### F.2 STOCHASTIC-D4RL MUJOCO SUITE

**Datasets** We adopt the *stochastic MuJoCo* protocol for risk-sensitive offline RL, following (Urpí et al., 2021). Policies are evaluated on

$$\{\text{HOPPER, WALKER2D, HALFCHEETAH}\} \times \{\text{MEDIUM-EXPERT, MEDIUM-REPLAY}\},$$

Compared to prior work, we prefer MEDIUM-EXPERT and MEDIUM-REPLAY to validate both *risk sensitivity* and *policy expressiveness* under multimodal action distributions. For training, we relabel per-transition rewards in the offline datasets to inject stochastic hazards (velocity or torso-pitch thresholds with Bernoulli penalties and early termination); *the same hazard model is used at evaluation.* All algorithms (CQL, CODAC, ORAAC, DiffusionQL/FlowQL, RADAC, etc.) are trained on these relabeled rewards; the hazard indicator is never provided as an input feature or mask, so no method receives privileged information about hazard locations. This ensures the critic and the policy are trained on the risk-aware rewards rather than only being tested under hazards.

**Settings** Each task defines a monitored signal and an additive Bernoulli penalty when a safety condition is violated; pose-based tasks also include an early-termination threshold.

- **HALFCHEETAH** : monitor forward velocity. Apply a penalty with probability $p = 0.05$ if the threshold is exceeded. Thresholds/penalties: MEDIUM-EXPERT/MEDIUM-REPLAY uses $v > 10.0$/ $v > 5.0$ with penalty $-70.0$. No early termination. Max episode steps: 200.

- **HOPPER / WALKER2D** : monitor torso pitch angle. When $|\theta|$ leaves the healthy range, add a penalty with probability $p = 0.10$; terminate early if $\theta > 2|\tilde{\theta}|$ . Max episode steps: 500.

  - HOPPER: healthy range $[-0.1, 0.1]$ rad; penalty $-50.0$ when $|\tilde{\theta}| > 0.1$; early termination if $|\theta| > 0.2$.
  - WALKER2D: healthy range $[-0.5, 0.5]$ rad; penalty $-30.0$ when $|\tilde{\theta}| > 0.5$; early termination if $|\theta| > 1.0$.

### F.3 BASELINES: IMPLEMENTATION & HYPERPARAMETERS

We include five representative offline-RL methods standard:

- **CODAC** (Ma et al., 2021) (distributional conservative learning). We primarily use the CVaR-optimizing specification ("CODAC-C", $\text{CVaR}_{0.1}$ objective). In all experiments, CODAC is configured with risk level $\alpha = 0.1$, so it serves as a "conservative + distributional CVaR" baseline against which RAMAC is compared.
- **ORAAC** (Urpí et al., 2021) (offline risk-averse actor–critic). A distributional critic with imitation-regularized policy optimizing a coherent risk objective.
- **CQL** (Kumar et al., 2020) (value pessimism). Non-distributional conservative Q-learning baseline. A hypothetical "CQL–CVaR" variant would combine CQL's conservative penalty with the same type of distributional CVaR head used in CODAC, making it conceptually very close to CODAC; we therefore treat CODAC as the representative conservative+CVaR baseline and do not report a separate CQL–CVaR instantiation.
- **DiffusionQL** (Wang et al., 2023) (expressive risk-neutral diffusion policy).
- **DiffusionQL** (Wang et al., 2023) (expressive risk-neutral diffusion policy).
- **FlowQL** (Park et al., 2025) (expressive risk-neutral flow-matching policy).

**Hyperparameter selection & tuning** For each of baselines, we run all baselines ourselves and tune the following parameters or adopt authors' recommended settings, mirroring the practice in Ma et al. (2021); Urpí et al. (2021); Wang et al. (2023); Park et al. (2025); Kumar et al. (2020).

- **FlowQL** (Park et al., 2025): we sweep the policy weight $\alpha \in \{1, 10, 30, 100, 1000\}$ per task and report the best-performing setting (selection by $\text{CVaR}_{0.1}$ unless noted).
- **DiffusionQL** (Wang et al., 2023): we consider $\eta \in \{0.1, 0.5, 1.0\}$ for BC coefficient . we use authors' recommended configuration for other parameters without retuning. We also used the best checkpoint of their model on each benchmark by following their protocol.
- **ORAAC** (Urpí et al., 2021): use the paper's recommended configuration (distributional critic, risk level $\alpha = 0.1$, anchor/prior regularization) without additional sweeps.
- **CODAC** (Ma et al., 2021): use the paper's tuned settings for D4RL (risk level $\alpha = 0.1$) without further tuning.
- **CQL** (Kumar et al., 2020): use the standard conservative coefficient and implementation defaults for MuJoCo locomotion.

### F.4 ESTIMATING OOD ACTION RATES AND DETECTORS

At evaluation time we measure the fraction of actions produced by a policy that fall outside the empirical action support of the offline dataset. Let $\mathcal{A}_{\mathcal{D}} = \{a_i\}_{i=1}^{N}$ denote the set of offline actions for a given task, and let $\mathcal{A}_{\text{eval}} = \{a_t^{(\text{eval})}\}_{t=1}^{T}$ be all actions emitted across evaluation rollouts (we use $S$ seeds and 10 episodes per seed; $S=5$ in the main table and $S=3$ in the detector-robustness ablation). Actions are already scaled to $[-1, 1]$ per dimension in MuJoCo, so we work directly in $\ell_2$ action space.

Given a detector that assigns an OOD indicator

$$\mathbf{1}_{\text{OOD}}(a_t^{(\text{eval})}) \in \{0, 1\},$$

we define the OOD action rate

$$\varepsilon_{\text{act}} \;=\; \frac{1}{T}\sum_{t=1}^{T}\mathbf{1}_{\text{OOD}}(a_t^{(\text{eval})}),$$

and report the mean and standard error over seeds. Because episodes may terminate early under the stochastic wrappers, $T$ is the actual number of executed timesteps, which makes rates comparable across seeds.

**1-NN detector (main text).** For each dataset action $a_i$ we compute its nearest neighbour among the other dataset actions,

$$d_i \;=\; \min_{j\neq i}\|a_i - a_j\|_2, \quad \text{medNN} \;=\; \text{median}\{d_i\}_{i=1}^{N},$$

and set the OOD threshold $\tau = \kappa \cdot \text{medNN}$ with $\kappa=3$. For each evaluation action,

$$d_t^{(\text{eval})} \;=\; \min_i \|a_t^{(\text{eval})} - a_i\|_2, \quad \mathbf{1}_{\text{OOD}}(a_t^{(\text{eval})}) = \mathbb{I}\{d_t^{(\text{eval})} > \tau\}.$$

Distances are computed efficiently via a KD–tree built on $\mathcal{A}_{\mathcal{D}}$. This 1–NN-based $\varepsilon_{\text{act}}$ is the quantity reported in Sec. 5.3.

**Alternative detectors.** To check that the RADAC < ORAAC trend is not an artifact of the 1–NN score, we also evaluate three density- and neighbourhood-based detectors on the same action space:

- **Gaussian KDE.** We fit a Gaussian kernel density estimator $p_{\text{KDE}}(a)$ to $\mathcal{A}_{\mathcal{D}}$ with bandwidth chosen by Scott's rule. For computational tractability, we subsample up to $5\times10^4$ dataset actions when estimating the threshold. Let $\ell(a) = \log p_{\text{KDE}}(a)$; we set $\tau$ to the $1\%$ lower quantile of $\{\ell(a_i)\}_i$ and mark an action as OOD if $\ell(a_t^{(\text{eval})}) < \tau$.

- **Local Outlier Factor (LOF).** We use the standard LOF score with 20 neighbours and contamination level 0.01. The detector is trained on $\mathcal{A}_{\mathcal{D}}$, and $\mathbf{1}_{\text{OOD}}(a_t^{(\text{eval})})=1$ when LOF predicts an outlier label for $a_t^{(\text{eval})}$.

- **Single-Gaussian Mahalanobis distance.** We fit a single multivariate Gaussian $\mathcal{N}(\mu,\Sigma)$ to $\mathcal{A}_{\mathcal{D}}$, with a small diagonal jitter added to $\Sigma$ for numerical stability. For any action $a$ we compute the squared Mahalanobis distance

$$d_{\text{Mah}}^2(a) \;=\; (a-\mu)^{\top}\Sigma^{-1}(a-\mu).$$

We estimate an upper-threshold $\tau$ as the $99\%$ quantile of $\{d_{\text{Mah}}^2(a_i)\}_i$ using up to $5\times10^4$ subsampled dataset actions, and declare $a_t^{(\text{eval})}$ OOD if $d_{\text{Mah}}^2(a_t^{(\text{eval})}) > \tau$.

For a seed-level rate $\hat{\varepsilon}$ computed by any of the detectors above, we approximate the standard error via a binomial model, $\text{SE} = \sqrt{\hat{\varepsilon}(1-\hat{\varepsilon})/T}$, and report the across-seed mean $\pm$ s.e.

Table 7: **RAMAC: hyperparameters.** We keep only the knobs that materially affect performance and stability. Values are our defaults; brackets show typical sweep ranges.

| **Global** | |
| --- | --- |
| Discount $\gamma$ | 0.99 |
| Batch size $B$ | 256 |
| Target update $\tau_{\text{target}}$ | 0.005 |
| Risk level $\alpha$ | 0.1 |
| **Critic (Deterministic IQN)** | |
| #Quantiles $N$ | 32 |
| Grid $\mathcal{T}_N$ | $\{(i - \frac{1}{2})/N\}_{i=1}^{N}$ (fixed) |
| Embedding dim | 128 |
| Critic LR | $3 \times 10^{-4}$ |
| Huber $\kappa$ | 1 (fixed) |
| Double IQN | enabled |
| **Actor (shared)** | |
| Actor LR | $3 \times 10^{-4}$ |
| BC weight $\lambda_{\text{BC}}$ | 1.0 |
| Risk weight $\eta$ | **RADAC**: 0.05 [0.02–0.1], **RAFMAC**: 1000 [100–1000] |
| Double critic clipping | **RADAC**:[150–150]-[300–300], **RAFMAC**:[300–300] |
| **RADAC-specific** | |
| Reverse diffusion steps $T$ | 5 (VP schedule) |
| **RAFMAC-specific** | |
| Flow steps $K$ | 10 (Euler, $\Delta t = 1/K$) |

