# OpenReview forum: "RAMAC: Multimodal Risk-Aware Offline Reinforcement Learning and Behavior Regularization"
_ICLR.cc/2026/Conference — ICLR 2026 Conference Desk Rejected Submission_

### Official Review · Reviewer_dseA · 2025-10-27

**Soundness:** 2
**Presentation:** 2
**Contribution:** 3
**Rating:** 2
**Confidence:** 4

**Summary:**

This paper investigates risk-aware offline reinforcement learning (RL). The authors employ a distributional critic with quantile regression to capture value distributions, which are then used to compute the Conditional Value at Risk (CVaR) metric. Unlike conventional methods that optimize for expected values, this approach leverages CVaR to guide policy optimization toward high-return regions while simultaneously controlling risk. To enhance the expressiveness of the policy class, the paper further adopts diffusion and flow-based policies, which can effectively model complex, multi-modal distributions and thereby improve the quality of behavior constraints. Experimental evaluations on a stochastic variant of the D4RL benchmark show that the proposed method can improve expected returns while also reducing risk.

**Strengths:**

1. Adopting a distributional critic to improve not only the expectation of returns but also the long-tail returns is important
2. The proposed idea is easy to follow and straightforward.
3. The proposed method is easy to be implemented, can be a simple baseline for future researches on this direction.

**Weaknesses:**

1. `Novelty` This paper seems to be a direct combination of 2 components: Diffusion Q Learning and distributional critic, which somehow lacks of novelty. Especially, this paper puts lots of efforts on Section 4 with 2 pages to emphasize the importance of utilizing expressive policy classes to conduct behavior regularization, but this has been widely studied in existing offline RL literatures, where using diffusion model or flow matching to model data distribution has become a popular choice.

2. `Soundness` The results in Figure 3 are strange. To the best of my knowledge, we can easily tune the conservatism strength in diffusion Q Learning or Flow Q Learning to achieve the similar behavior of RADAC. So, Figure 3 (c, d) looks like lacking of hyperparameter tuning. Also, it seems that Figure3 (f-g) produces lots of OOD actions, which can be easily addressed through a small scale of pertubation. Considering the strange results in Figure 3, I cannot gaurantee the soundness of this paper.

3. Figure 4 can somehow demonstrate that RADAC can produce safe behaviors, but I cannot see why. The objective of RADAC is to maximize the CVaR of `REWARD` and imposes no gaurantee of `safty`. It would be better to present the return distributions of the rollouted trajectories, which can directly reflect the training results of RADAC objective.

4. `Theory` The authors take theories as a core contribution of this paper, but the theoretical insights have been widely studied in existing offline RL literature. For example, utilizing expressive policies can produce better results than unimodal policies.

**Questions:**

Please see weaknesses for details.

---

> ### Author Response · Authors · 2025-11-22
>
> We thank the reviewer for the detailed feedback. We address the main points on (i) novelty, (ii) soundness of the toy experiment and hyperparameters, (iii) the interpretation of Fig. 4 and “safety”, and (iv) the scope of the theory.
>
>   1.Novelty (distributional critic + expressive policy + CVaR):
>
> We agree that diffusion/flow-based expressive policies and distributional critics are established building blocks, and that the current draft does not clearly separate what is new. Our contribution is to make these expressive actors explicitly risk-awareby coupling a distributional critic with a CVaR objective and backpropagating tail-risk signals through the full generative trajectory,while a behavior-cloning regularizer controls OOD actions via a TV–KL argument which is, to our knowledge, has not been made explicit in prior risk-aware offline RL with behavior regularization. We are revising the Introduction and Sec. 2–4 to to reflect this positioning.
>
>    2. Soundness of the toy experiment and hyperparameters:
>
> The 2-D risky bandit is constructed so that the behavior data lie on a thin ring around a safe center; our intention is not to claim that anchor–perturbation methods or DiffusionQL/FlowQL cannot be tuned differently, but to highlight how they interact with the data support. As discussed in Sec. 4.1, for thin or non-convex supports such as the ring, any ε-ball around a single behavior anchor will eventually intersect off-support once the local margin m(b) becomes small, and the learned residual is not constrained to stay tangent to the data manifold; even small perturbations can thus systematically produce OOD actions, which is what Fig. 3(f–g) is meant to illustrate. For DiffusionQL/FlowQL, we use the hyperparameter grids documented in App. E/G. Increasing conservatism reduces OOD mass and hazard visits but leaves the policies risk-neutral, so they continue to chase high-Q yet risky regions; decreasing the RL weight recovers BC-like behavior, as shown in App. F.1.
>
> We are currently running an explicit toy-bandit hyperparameter sweep varying the BC/RL weights and CVaR α, and we will add the resulting curves (mean, CVaR, and OOD/hazard counts) in the appendix to show how RADAC and the baselines behave across reasonable tuning ranges.
>
>    3. Fig. 4 and the notion of “safety”:
>
> We agree that our use of the word “safe” can be misleading, and that plotting only final policies does not fully justify this term. We do not claim formal safety guarantees. In Sec. 5.2 we instead operationalize “safety” as avoiding hazard events (e.g., entering designated risky regions, high-velocity collisions, early termination) under the stochastic-D4RL hazard model; App. G.2 specifies these hazards in detail. Under this definition, RAMAC reallocates mass away from hazard regions while still targeting high-return modes, which is reflected in lower hazard counts and improved CVaR in Table 1 and Sec. 5.3.
>
> In the revision we are (i) moving a concise summary of the hazard model from App. G.2 into Sec. 5, and (ii) adding simple plots/tables of empirical return distributions and CVaR quantiles together with hazard rates so that readers can directly see how RADAC reshapes tail outcomes.
>
>    4. Scope of the theory:
>
> We agree that broad statements like “expressive policies help” are well known, and we did not intend our unimodal vs. multimodal conjecture to be read as a major theoretical contribution. In the revision, we narrow the formal focus to two observations directly tied to RAMAC: (i) a per-state OOD bound for behavior-regularized generative policies obtained via a TV–KL inequality, showing that minimizing the BC/MLE term reduces the probability of sampling actions outside the dataset support, and (ii) a geometric explanation of why prior-anchored (anchor–perturbation) policies on thin or non-convex supports can still assign mass to OOD regions between disjoint modes, even when a BC prior is used. We are rewriting Sec. 4 to clearly connect these points and to avoid over-claiming general theorems about all expressive policies.

---

> > ### Author Response · Authors · 2025-12-03
> >
> > We thank the reviewer for the detailed and thoughtful feedback. Below we address each concern and summarize the corresponding changes in the revised manuscript.
> >
> > **(1) Novelty: beyond “distributional critic + diffusion/flow”.**
> > We agree that both distributional critics and expressive diffusion/flow policies are now standard building blocks, and that the original draft did not make our contribution sufficiently explicit. In the revision, Sections 1–3 clarify that RAMAC is not merely stacking a distributional critic onto DiffusionQL or FlowQL, but (i) **training a single expressive generative actor directly on a composite BC + CVaR objective** and (ii) **analyzing how the BC term controls per-state out-of-distribution (OOD) mass**. Concretely, whereas prior diffusion/flow methods with distributional critics still update the actor via an expectation-based objective (for example, gradients of E[Q]), RAMAC differentiates through CVaR at level alpha of the return distribution Z(s, a) itself, so the actor receives tail-sensitive gradients, and the same forward-KL BC term that fits the data distribution also yields an explicit upper bound on the OOD mass in Section 4.2. To our knowledge, this is the first offline risk-aware framework that (a) couples a diffusion or flow actor with a **direct CVaR objective** and (b) links its BC regularizer to a **per-state OOD bound** specialized to expressive generative policies. We also clarify the relation to ORAAC- and CQL-style risk-aware baselines and to mixture-policy CVaR methods in the related work, highlighting that those operate with parametric or tabular actors or conservative penalties rather than a BC + CVaR trained expressive generator.
> >
> > **(2) Soundness of the toy experiment and Fig. 3.**
> > We agree that, without more context, Fig. 3 can look like a hyperparameter artifact. In the revision we therefore make the **geometry and tuning** of the Toy Risky Bandit explicit (Section 4.3 and Appendix E.1). The behavior distribution consists of a **safe center** (moderate mean, light tail) and a **risky outer ring** (higher mean, heavy lower tail). We first show in a new figure that **BC-only diffusion and flow policies faithfully reconstruct the bimodal behavior distribution**, confirming that the actors are expressive enough to capture both modes and that the final concentration at the center under RADAC is not due to representational failure.
> >
> > To address the tuning concern, Appendix E.2 and E.8 now includes **sweeps over the RL/BC weights and the CVaR level** for DiffusionQL, FlowQL, ORAAC-style anchor–perturbation, and RADAC, reporting the mean, CVaR at level 0.1, and OOD or hazard counts across a wide range of conservative settings. Increasing conservatism in DiffusionQL and FlowQL indeed reduces OOD mass and hazard visits, but we observe a structural trade-off: either they (a) remain risk-neutral and keep assigning probability to the high-mean yet hazardous ring, or (b) effectively revert to behavior cloning and collapse away from the risky mode, without matching RADAC’s combination of strong tail performance and high mean. For prior-anchored perturbation methods, we add a simple lemma (Section 4.1, Appendix B.1) formalizing the intuition behind Fig. 3(f–g): on thin or non-convex supports such as the ring, once the anchor ball intersects off-support regions, any nonzero perturbation radius yields a strictly positive per-state OOD probability, so even “small” perturbations do not eliminate leakage into hazardous areas. We will further clarify this in the text to make it clear that Fig. 3 is meant to illustrate this geometric limitation rather than poor tuning.
> >
> > **(3) Fig. 4, “safety”, and rollout return distributions.**
> > We agree that our original use of the term “safe” around Fig. 4 was ambiguous. In the revision we explicitly state that we **do not claim formal safety guarantees**. Instead, Section 5.2 defines an operational notion of “safety” tied to the stochastic hazard wrapper: a trajectory is “unsafe” when it triggers hazard events (entering designated risky regions, high-velocity collisions, early termination, and so on), implemented as large negative rewards plus truncation. A concise description of this hazard model is now moved from the appendix into Section 5.2 so that the reader can see exactly what Fig. 4 is visualizing.

---

> > > ### Author Response · Authors · 2025-12-03
> > >
> > > We also follow the reviewer’s suggestion and **add an analysis of rollout return distributions**.  Appendix E.7 (return-distribution plots) contains detailed figures. For each method and environment, we aggregate 30–60 evaluation rollouts across three seeds and plot histograms or kernel density estimates of episodic returns, together with vertical lines for the mean, median, and CVaR at level 0.1. Because hazards correspond to large negative rewards and often early termination, catastrophic episodes populate the extreme left tail. On `halfcheetah-medium-expert-v2`, for example, RADAC concentrates almost all mass in a high-return band with CVaR at level 0.1 around 4.7 × 10^3, while FlowQL has a similar mean but a noticeably heavier left tail (CVaR at level 0.1 around 3.7 × 10^3), and ORAAC exhibits a multimodal distribution with occasional near-zero or negative returns (CVaR at level 0.1 around 7.4 × 10^2). DiffusionQL and CQL largely collapse to near-zero or negative returns. Together with the hazard counts in Table 1, these distributions show that RADAC reduces the probability mass in the catastrophic tail while preserving high-return modes, whereas risk-neutral expressive baselines either keep a heavier catastrophic tail or avoid hazards only by collapsing. We also soften the wording around Fig. 4 and explicitly point readers to Section 5.3 and Appendix E.7 for the quantitative rollout distributions.
> > >
> > > **(4) Theory: scope and what is actually new.**
> > > We appreciate the concern that statements like “expressive policies help” are already well known. In the revision we **narrow the theoretical scope** and avoid presenting such high-level intuitions as core contributions. Section 4 now focuses on two concrete, RAMAC-specific points:
> > > (i) a **geometric analysis of prior-anchored perturbation** on thin or non-convex supports, making precise why anchor-based methods can maintain nonzero OOD mass even when the anchor is behavior-like, and
> > > (ii) a **BC-regularized generative-actor bound**, which uses a TV–KL inequality to show that minimizing the forward-KL BC loss directly controls the per-state OOD mass and extends this to a simple CVaR degradation bound under bounded rewards.
> > >
> > > We position these as **modest but specific** theoretical observations that (a) explain the behavior seen in the toy and Stochastic-D4RL experiments and (b) clarify how RAMAC’s BC + CVaR objective controls both OOD actions and lower-tail risk, rather than as broad new theorems about expressive policies in general.
> > >
> > > We hope these clarifications and additions address the reviewer’s concerns on novelty, soundness, and theory, and we appreciate the suggestion that RAMAC can serve as a simple baseline for future work in risk-aware offline RL.

---

### Official Review · Reviewer_ktys · 2025-10-28

**Soundness:** 2
**Presentation:** 2
**Contribution:** 3
**Rating:** 2
**Confidence:** 4

**Summary:**

This paper presents RAMAC for offline risk-aware RL.  The RAMAC methodology combines an expressive generative actor with a distributional critic.  The actor-critic approach is trained to optimize CVaR for tail-based risk-sensitivity.  The loss function avoids out-of-distribution actions by incorporating a regularizer that encourages behavior cloning.  Experiments on stochastic D4RL tasks show improved CVaR in many settings, while simultaneously maintaining high average returns.

**Strengths:**

One of the biggest strengths of this paper is the empirical evaluation in Sec. 5.  The authors chose a strong set of baselines to compare with, and report, both, CVaR and average return across challenging instances.  The authors provide good insights on the behavior of OOD action selection, and provide some theoretical analysis that justifies why their method works well.

**Weaknesses:**

Overall, my opinion is that this paper is not quite ready for publication in its current form.  The presentation requires a bit of polishing.  Indeed, the paper briefly covers many concepts without adequate and thorough discussion.  These include concepts of offline RL, risk-sensitive decision making, OOD actions, distributional RL, behavior regularization, and one of the most terse descriptions of diffusion methods that I have come across.

In some ways RAFMAC feels like a bit of an afterthought in the way that it is presented.  There is almost no discussion of RAFMAC throughout the paper, with most attention being paid to RADAC.  The example result in Fig. 3 only evaluated RADAC (I realize that there is some evaluation of RAFMAC in the appendix).  Only Table 1 evaluates RAFMAC in any capacity, and frankly the results are not compelling for it.  To what extent should we evaluate RAFMAC as a core contribution of the paper?  If it is not intended to be a core contribution, and indeed doesn't perform that well, then perhaps it is better to relegate it to the appendix?

The authors claim that "theoretical insight" is a core contribution of this work.  However, the analysis amounts to a single Proposition and a conjecture, which is a bit limited as far as theoretical analysis is concerned.  In terms of the proposition, I am a bit unclear on the implications of this result, and in my mind the authors don't adequately explain.  The conjecture, on the other hand, is quite obvious--namely, that a multimodal policy should be more flexible (in terms of lower KL) than a unimodal policy.  Note that the authors don't properly define either of these policies.

The experimental setup is a bit unclear.  First, it is not clear how the 2-D contextual bandit experiment in Sec. 4.3 is designed.  From Fig. 3 it is unclear to the reader whether the RADAC result is desireable or not.  In fact, the authors state that "RADAC concentrates near the safe center without losing multimodality," but by failing to capture the outer ring I would argue that it has failed to capture multimodality.  For Sec. 5 the description of the environments is somewhat lacking.  I am unclear on the risk signal in these environments without diving deeper into Fu et al. (2020).  Looking at the qualitative distribution plots in Fig. 4 I would argue that there is limited signal that RAMAC learns to avoid unsafe regions relative to baselines.

Finally, because there are so many components to RAMAC, the authors should include a careful ablation study in the main text.  There are limited ablation studies in the appendix, but I think these need to be strengthened and included in the main text.

Minor issues:
* In the last paragraph on Page 1 you have double parenthesis on many references.
* L139-140 : I am not sure what the following statement actually means "our framework instead inject distributional risk signals."
* The behavior cloning loss $\mathcal{L}_{BC}$ is not explicitly defined

**Questions:**

See Weaknesses section above.

---

> ### Author Response · Authors · 2025-11-22
>
> We thank the reviewer for the constructive comments on presentation, the role of RAFMAC, the theory claims, and the clarity of the toy and Stochastic-D4RL setups. We respond point by point.
>
>    1. Presentation and conceptual coverage:
>
> We are revising Sec. 2 (Preliminaries) and the first part of Sec. 3 to add a clearer high-level overview: (i) a short explanation of offline RL and of the main families of offline RL methods, (ii) an explicit discussion of behavior regularization and why it remains competitive in recent work (e.g., BRAC, follow-ups such as Tarasov et al. 2023), and (iii) a more intuitive introduction to distributional critics and CVaR before presenting equations. These edits are purely expository and do not change the method or results.
>
>    2. Role of RAFMAC:
>
> We agree that, in the current draft, RAFMAC should not be presented as a core contribution. Our main goal is to study CVaR-through-generation under behavior cloning with an expressive generative actor, and for this we focus our analysis and tuning on the diffusion-based instantiation (RADAC). RAFMAC is a straightforward flow-matching variant that uses the same L_π = L_BC + η L_CVaR objective and did not show additional conceptual or empirical benefits beyond what RADAC already demonstrates as you have pointed out.
> In the revision, we are (i) updating the Introduction and Sec. 3 to present RADAC as the primary instantiation of RAMAC, (ii) moving the flow-matching variant (RAFMAC) to the appendix as a complementary example illustrating that the same BC + CVaR objective also applies to flow-based policies, and (iii) adjusting the main-text tables and discussion to avoid over-emphasizing RAFMAC’s weaker results (keeping any RAFMAC numbers, if retained, clearly labeled as supplementary). This keeps the narrative focused on RADAC while still allowing interested readers to inspect the flow-matching variant.
>
>    3. Theory:
>
> We agree that broad statements like “expressive policies help” are well known, and we did not intend our unimodal vs. multimodal conjecture to be read as a major theoretical contribution. In the revision, we will soften the wording in the Introduction/Contributions  and we are either removing this conjecture from the main text or moving it to an intuition-style discussion instead of presenting it next to formal results. Our theoretical focus is narrowed to two concrete observations directly tied to RAMAC. First, we clarify a per-state OOD bound for behavior-regularized generative policies via a simple TV–KL inequality, explicitly showing that minimizing the BC/MLE term reduces, for each state, the probability of sampling actions outside the dataset support. Second, we sharpen the geometric discussion of prior-anchored (anchor–perturbation) policies on thin or non-convex supports: even when a BC prior is used, constraining the policy to small perturbations around a single anchor can still place non-trivial mass in OOD regions that lie between disjoint modes, which explains why such methods can violate the per-state OOD control intuition that holds for truly generative BC-style actors. We are rewriting Sec. 4 to connect these points and to avoid over-claiming general theorems about expressive policies.
>
>   4.Toy design, environment clarity, and Fig. 4:
>
> The contextual bandit in Sec. 4.3 is constructed so that the ground-truth action distribution has a safe centerand a risky ring . As we show in App. F.1, a BC-only expressive generator faithfully reproduces both modes, confirming that the model can represent the multimodal data. Under RADAC, the CVaR term gradually shifts probability from the risky ring to the safe center while BC keeps samples on-manifold; the final concentration at the center is therefore a consequence of risk optimization, not an inability to represent the outer mode. This toy settings and results are  analogous to ( Wang et al. 2023) toy experiment. In the revision we will make this intent explicit in Sec. 4.3 and in the Fig. 3/4 captions.
>
>
> For the Stochastic-D4RL experiments, we are adding a concise summary at the beginning of Sec. 5 that explains the construction of the risky environments (how hazards are injected, what penalties are used, and how this differs from the standard D4RL benchmarks), and we point to an expanded description and visualizations in the appendix. This should make it much easier to interpret our results as showing that RAMAC learns to avoid unsafe regions relative to baselines.
>
>    5. Minor issues:
> We will fix the double parentheses in the last paragraph of Page 1 in the revision. We will also rephrase the sentence around L139–140 to clarify what it means for “our framework to inject distributional risk signals” (namely, that the actor receives gradients from a distributional critic under a CVaR objective, not only scalar Q-values). Finally, we will explicitly define the behavior cloning loss L_bc in Sec. 3 when first introduced.

---

> > ### Author Response · Authors · 2025-12-03
> >
> > We thank the reviewer for the detailed and constructive feedback. Below we explain how the revision addresses the concerns on presentation, the role of RAFMAC, theory, the toy/Stochastic-D4RL setups, and ablations.
> >
> > **(1) Presentation and conceptual coverage.**
> > We  revised Sec. 2 (Preliminaries) and the early part of Sec. 3 to clarify the conceptual pipeline:
> > (i) **Offline RL and OOD.** We now explicitly define offline RL, the dataset support $\mathrm{supp}(\mathcal{D})$, and the notion of OOD actions, and explain why OOD visitation is a key driver of catastrophic failures (new first paragraph of Sec. 2, highlighted in blue in the revision).
> > (ii) **Behavior regularization.** We added a BRAC-style actor–critic paragraph in Sec. 2 that reviews behavior-regularized objectives, and we explicitly connect RAMAC’s BC term to this pattern in Sec. 3.3.
> > (iii) **Distributional RL and CVaR.** Sec. 2 now introduces distributional critics and CVaR in a self-contained way before any equations are used (“Distributional RL and Risk Measures”), making the risk objective easier to follow.
> > (iv) **Diffusion/flow policies.** We expanded the description of expressive generative policies, clarifying how diffusion and flow-matching policies define differentiable trajectories $a = \psi_\theta(s, z)$ and how RAMAC uses them (“Expressive Generative Policies as Differentiable Trajectories”). The previously terse sentence that “our framework injects distributional risk signals” has been replaced by a precise explanation that the actor receives gradients of tail-sensitive quantities such as $\mathrm{CVaR}_\alpha(Z^\pi(s,a))$ from the distributional critic and backpropagates them through the diffusion/flow trajectory.
> >
> > We also now explicitly define the behavior cloning loss
> > $\mathcal{L}*{\mathrm{BC}}(\theta) = -\mathbb{E}*{(s,a)\sim\mathcal{D}}[\log \pi_\theta(a \mid s)]$
> > when it is first introduced in Sec. 3.3, and we fixed the double parentheses in the last paragraph of page 1.
> >
> > **(2) Role of RAFMAC.**
> > We agree that in the original submission RAFMAC appeared underdeveloped relative to RADAC. In the revision we therefore de-emphasize RAFMAC as a core contribution and focus the narrative on the diffusion-based instantiation:
> > (i) Sec. 1 and Sec. 3 now present RADAC (diffusion actor) as the primary instantiation of RAMAC, and describe RAFMAC only as a secondary flow-matching variant.
> > (ii) All main-text tables now report RADAC as the sole RAMAC instantiation; RAFMAC results have been moved to the Appendix E.2, where they are clearly labeled as supplementary.
> > (iii) The text in Sec. 5 has been updated to explicitly state that RAFMAC is reported in the Appendix only “as a complementary example illustrating that the same BC+CVaR objective also applies to flow-based policies.”
> > This makes RADAC the clear focal point of the empirical study, while still allowing interested readers to inspect the flow-based variant.
> >
> > **(3) Theory: scope and implications.**
> > We have narrowed and clarified our theoretical claims:
> > (i) The informal “unimodal vs. multimodal” conjecture is no longer presented as a core contribution; we removed it from the main claims and only refer to it as intuition where relevant.
> > (ii) Sec. 4 has been rewritten as “Behavior Regularization in Offline RL,” focusing on two concrete points directly tied to RAMAC:
> > (a) **Prior-anchored limitations.** We added a lemma and a geometric discussion (with proof in the Appendix B.1) showing that, when the anchor ball overlaps off-support regions, prior-anchored perturbation schemes inevitably maintain a strictly positive per-state OOD probability, especially on thin or non-convex supports. This formalizes the “leakage” intuition in a simple but explicit way.
> > (b) **BC-regularized generative actors.** A proposition (now highlighted and explained in Sec. 4.2) shows that, for expressive generative policies trained with BC on the deployed actor, the forward KL $D_{\mathrm{KL}}(\beta ,\Vert, \pi_\theta)$ yields an explicit upper bound on the per-state OOD mass $\delta_s(\pi_\theta)$. We also extend this to a simple CVaR degradation bound under bounded rewards, clarifying how behavior regularization controls both OOD and lower-tail risk. To the best of our knowledge, this is the first objective-level per-state OOD bound specialized to BC-regularized expressive policies, and the revision makes this contribution explicit in the abstract, introduction, Sec. 4, and conclusions.

---

> > > ### Author Response · Authors · 2025-12-03
> > >
> > > **(4) Toy design, environment clarity, and Fig. 4.**
> > > For the 2-D contextual bandit (Toy Risky Bandit), we now make the design and intended takeaway precise:
> > > (i) Sec. 4.3 and the corresponding figure now explain that the ground-truth distribution consists of a *safe center* (moderate mean, light tail) and a *risky ring* (higher mean, heavy lower tail). We emphasize that the diffusion/flow generators are expressive enough to capture both modes: the Appendix E.2 (new figure) shows BC-only diffusion/flow policies faithfully recovering the multimodal behavior distribution.
> > > (ii) The Appendix E.2 further adds epoch-by-epoch RADAC snapshots, showing that RADAC first matches the bimodal data under BC and then gradually shifts mass from the risky ring to the safe center under the CVaR-guided loss. We explicitly clarify in the text that the final concentration at the center is a consequence of risk optimization rather than a failure to represent the outer mode.
> > > (iii) For Stochastic-D4RL, we added a concise description of the risk construction at the beginning of Sec. 5: how hazards are tied to forward velocity or torso angle, how rare heavy-tailed penalties and early termination are injected, and how rewards are relabeled. In Sec. 5.2 we also clarify the interpretation of the safety plots (Fig. 4), emphasizing how RADAC concentrates mass inside hazard-free bands while DiffusionQL and ORAAC either collapse or leak into risky regions. We also add Return distribution of trajectories from RADAC, Diffusion QL, ORAAC, FLow QL, CQL. This helps to answer your question that the RADAC received meaningful information with its core structure as shown in qualitative plots of Fig.4.
> > >
> > > **(5) Ablations and component analysis.**
> > > In response to the request for stronger ablations in the main text, we have:
> > > (i) Added an explicit OOD-action analysis section (Sec. 5.3), where we measure the $\varepsilon_{\mathrm{act}}$ rate using 1-NN distance and show that RADAC consistently has lower OOD mass than ORAAC across tasks (table in the main text), with additional detectors in the Appendix E.4. This directly ties back to the theoretical behavior-regularization results.
> > > (ii) Expanded the toy experiments (Appendix E.2) with sweeps over the RL weight $\eta$ for DiffusionQL/FlowQL and RADAC, and over the risk weight $\lambda$ for ORAAC-style methods, illustrating how risk-neutral expressive baselines chase high-mean but hazardous regions while ORAAC-style methods hold their structural limitations claimed in Sec 4.1,  and how RADAC trades off ring coverage for tail-risk reduction while staying on-manifold.
> > > (iii) Retained the distortion and hyperparameter analyses (e.g., CVaR vs. alternative distortions, $K$/$N$ choices, and Alpha sweep) in the Appendix E.6 & 8, but we now reference them explicitly from Sec. 5 so that readers see how each component (BC, CVaR, expressive generator) contributes.
> > >
> > > We hope these revisions address the reviewer’s concerns and clarify both the conceptual and empirical contributions of RAMAC.

---

### Official Review · Reviewer_JUg6 · 2025-10-30

**Soundness:** 2
**Presentation:** 3
**Contribution:** 3
**Rating:** 4
**Confidence:** 3

**Summary:**

This paper tackles offline RL for safety-critical settings where one must learn purely from logged data but still avoid “catastrophic” outcomes. The authors propose RAMAC (Risk-Aware Multimodal Actor-Critic): pair an expressive, generative policy (either a diffusion model or a flow-matching model) with a distributional critic that models the full spread of possible returns instead of just the average. They then optimize a combined objective that (a) behavior-clones the dataset to stay “on-manifold” and (b) pushes the policy away from the lower tail of the return distribution by maximizing CVaR (a standard risk measure). This way, the policy remains expressive enough to capture multimodal behaviors but becomes explicitly risk-aware. Across Stochastic-D4RL benchmarks, the two instantiations generally improve upon baselines.

**Strengths:**

I think the topic itself is important. I like that the paper targets risk in offline RL, because it seems directly relevant to safety-critical applications where exploration is impossible.

The idea of threading CVaR gradients through the full generative path of diffusion/flow actors seems a clean way to marry expressiveness with risk shaping.

It is interesting to see the Stochastic-D4RL study and the OOD-action analysis, it seems the methods improve CVaR on several tasks and show lower OOD rates than prior-anchor methods.

**Weaknesses:**

On the diffusion/flow side, can you clarify how your approach differs from concurrent “distributional critic + diffusion” works that still optimize expectation—what concrete failure case of those methods does RAMAC fix (beyond the conceptual argument)?

It seems there is a missing related work. “A Simple Mixture Policy Parameterization for Improving Sample Efficiency of CVaR Optimization” by Luo et al. proposes a mixture-policy parameterization specifically to improve CVaR optimization efficiency by blending a risk-neutral and an adjustable policy to shape the lower tail. The risk-neutral part is trained by offline RL. It’s squarely in your problem space, even if their mechanism (mixture policy for CVaR) differs from yours (distributional critic + generative actor).

Some more experiments:

First critical thing. Your theory motivates expressive actors for lower forward-KL on multimodal data; can you test an explicit-likelihood actor (e.g., CVAE with improved decoding) under the same BC+CVaR objective to isolate whether the gains truly come from trajectory-based generation vs. expressiveness alone?

For fairness, you relabel rewards with the same hazard model used in evaluation; could you expand on potential label leakage risks and justify why this procedure does not advantage RAMAC versus baselines? A short sensitivity study to hazard severity would strengthen the claim.

could you add CODAC-CVAR and CQL-CVAR variants tuned specifically for lower-tail metrics, and also include high-confidence policy improvement (e.g., SPIBB-style) to show how RAMAC compares to explicit safety-first policies? This would better cover the design space you survey.

Your results currently fix α=0.1; a risk-return frontier (vary α) would make the safety/return trade-off more transparent and would help practitioners pick α. I think this sensitivity should be important for risk-averse RL.

It is known that diffusion backprop through denoising steps can be heavy; it would be interesting to report computation time, and inference latency (vs. Flow) to guide adoption, and consider your suggested distilled one-step head as an ablation.

OOD detection robustness: the 1-NN detector is simple and sensible; could you replicate with kernel density or ensemble-based novelty measures to ensure the OOD trend (RADAC < ORAAC) is detector-agnostic?

**Questions:**

see above weaknesses. I will adjust my score depending on the author response.

---

> ### Author Response · Authors · 2025-11-21
>
> We thank the reviewer for the detailed and constructive comments. We address the main points and indicate the revisions and additional experiments we are currently working on.
>
>   1. Difference from distributional-critic + diffusion:
>
> DiffusionQL/FlowQL with IQN still optimize expected return; the distributional critic improves value estimates but tail risk only affects the policy indirectly. RAMAC instead optimizes a BC + CVaR objective: CVaR gradients from the distributional critic are backpropagated through the generative trajectory while BC keeps actions on-manifold, so the actor directly shifts probability mass away from low-quantile regions. This explains the gaps in Fig. 3 and Table 1: in the Toy Risky Bandit and hazardous control tasks RAMAC attains higher CVaR_0.1 at similar mean return and lower hazard rate than DiffusionQL/FlowQL. We will clarify this contrast in Sec. 1/Sec. 2.
>
>    2. Relation to Luo et al. and explicit-likelihood actors:
>
> Thanks for suggesting this paper. We will add Luo et al. (“A Simple Mixture Policy for CVaR-Optimized RL”) as related work. Their method mixes risk-neutral and risk-averse policies under a standard parametric actor, whereas RAMAC uses a single expressive generative actor with a distributional critic, explicit BC + CVaR + per-state OOD constraints, and targets offline risk-sensitive control with hazard relabeling.
> We agree that testing an explicit-likelihood actor would more cleanly separate “trajectory-based generation” from “expressiveness alone.” In this paper, we focus our empirical study on two expressive generative instantiations (diffusion and flow) and will soften the theoretical discussion of multimodal vs. unimodal policies accordingly in the revision.Importantly, the BC + CVaR objective and the per-state OOD bound in Sec. 4 apply equally to explicit-likelihood actors such as CVAEs (with Lbc=−logπθ(a∣s)) and we will clarify this. A careful implementation and tuning of a CVAE-based actor-critic across all tasks would require substantial additional engineering, so if time permits during the discussion phase and we obtain stable pilot results with a CVAE-based actor on a representative task, we will include them as an additional ablation in the appendix.
>
>    3. Hazard relabeling and severity:
>
> All methods (RAMAC, ORAAC, DiffusionQL/FlowQL, CODAC, etc.) are trained and evaluated on the same hazard-relabeled datasets; the hazard model is never given explicitly to any agent, so RAMAC does not receive extra information. We are currently running a small sensitivity study where we vary hazard penalty magnitude/frequency. For each setting we will report mean, CVaR_0.1 and hazard rate and show that the ranking between RAMAC and strong baselines is stable. We will also describe the relabeling procedure more clearly.
>
>    4. Baselines: CODAC-CVaR / CQL-CVaR / SPIBB:
>
> Our CODAC baseline already uses the CVaR-optimizing specification from Ma et al. (α = 0.1); we will state this explicitly in our revision. Constructing “CQL–CVaR” would require a distributional critic and CVaR objective that are conceptually close to CODAC’s conservative + CVaR design, so we do not expect it to change our qualitative conclusions. SPIBB-style high-confidence policy improvement is formulated for discrete/tabular settings; adapting it to continuous control would require additional machinery beyond our scope. We will add a discussion that such high-confidence methods are complementary to our approach and that combining them with expressive generative actors is interesting future work.
>
>    5. Risk–return frontier over α:
>
> We agree that varying the risk level is important. We are currently training RAMAC with several α values on representative tasks. In the revision we will report mean and CVaR_α for each α and plot a risk–return frontier, commenting on how its shape depends on hazard structure.
>
>    6. Runtime, latency, and distilled head:
>
> We are currently running a small runtime study that reports training wall-clock time, and action inference latency for RAMAC vs DiffusionQL/FlowQL under identical hardware and budgets. We will include these results in the revision. We also agree that a distilled one-step head is a promising way to further reduce inference latency. We plan to explore it in follow-up work. If time permits and we obtain stable results for a distilled head on at least one representative task during the discussion phase, we will report them as an additional ablation in the revised manuscript.
>
>    7. OOD detection robustness:
> To check robustness, we are currently evaluating alternative detectors (e.g., KDE and LOF-style local outlier scores) on the same feature space for representative tasks. In the revision we will report OOD counts and RADAC/ORAAC rankings under these detectors.

---

> > ### Author Response · Authors · 2025-12-03
> >
> > We thank the reviewer for the helpful feedback. Below we address the main concerns and summarize the changes in the revision.
> >
> > **(1) Difference from distributional-critic + diffusion/flow.**
> > In the revision we make the novelty relative to concurrent diffusion/flow works more explicit in the abstract, Introduction (Sec. 1) and Preliminaries (Sec. 2). Closest concurrent methods pair diffusion policies with distributional critics, but still train the actor on an expectation-based objective: the critic models the return distribution while the policy update depends on $\mathbb{E}[Q]$, so lower-tail outcomes only influence the actor indirectly. By contrast, RAMAC couples an expressive generative actor with a distributional critic and directly differentiates a composite BC+CVaR objective (Eq. (12)): gradients of the bottom-$\alpha$ quantiles are backpropagated through the full diffusion/flow trajectory, while the BC term induces a forward-KL upper bound on the per-state OOD mass $\delta_s(\pi_\theta)$ (Sec. 4).
> >
> > **(2) Relation to Luo et al.**
> > We cite and discuss Luo et al.  (“A Simple Mixture Policy for CVaR-Optimized RL”) as related work in Appendix C（**Mixture-Policy CVaR Optimization**). Their method blends a risk-neutral and an adjustable policy under a standard parametric actor trained online, whereas RAMAC targets offline control with (i) a single expressive generative actor, (ii) a distributional critic, and (iii) a BC+CVaR objective with per-state OOD control.
> >
> > **(3) Hazard relabeling and severity.**
> > In our revision, Sec. 5 and Appendix F.2 now describe the hazard-relabeled rewards in detail. We treat the hazard model as part of the environment’s reward and termination: all methods (CQL, CODAC, ORAAC, DiffusionQL/FlowQL, RAMAC, etc.) are trained on the same relabeled transitions, and the hazard indicator is never provided as an input feature or mask, so no method receives privileged information about hazard locations.
> >
> > **(4) Additional baselines (CODAC--CVaR, CQL--CVaR, SPIBB-style).**
> > We now state explicitly that our CODAC configuration uses the CVaR-optimizing specification of (Ma et al). with risk level $\alpha = 0.1$, so it already serves as a “conservative + distributional CVaR’’ baseline provided in Appendix F.3. We also add a short discussion noting that a hypothetical “CQL-CVaR’’ would combine CQL’s conservative penalty with a distributional CVaR head and is therefore conceptually very close to CODAC, so we do not expect it to change the qualitative picture. Finally, Limitation Section, Appendix. A clarifies that SPIBB-style high-confidence policy improvement is currently formulated for discrete/tabular settings, and we highlight combining such high-confidence constraints with expressive generative actors as an interesting direction for future work.

---

> > > ### Author Response · Authors · 2025-12-03
> > >
> > > **(5) Risk–return frontier over $\alpha$.**
> > > We agree that varying the risk level is important in practice. In the revision we add a dedicated analysis of the risk–return frontier in Appendix E.8 (Fig.14). For RADAC on two representative tasks, halfcheetah-medium-replay-v2 and walker2d-medium-replay-v2, we train with $\alpha \in {0.05, 0.10, 0.20}$ and report the seed-averaged mean normalized score versus $\mathrm{CVaR}*\alpha$, with error bars indicating the standard error across seeds. On halfcheetah-medium-replay-v2, changing α mainly affects the lower tail: the CVaR value moves substantially while the mean remains in a tight band, showing that RADAC can reshape tail risk with minimal impact on average performance. On walker2d-medium-replay-v2, more tail-sensitive settings improve both CVaR and the mean simultaneously, indicating that suppressing catastrophic low-return trajectories can also raise average returns. Overall, this frontier provides practitioners with a simple knob to trade off tail risk and mean performance when choosing α.
> > >
> > > **(6) Runtime, latency, and distilled heads.**
> > > Appendix. E.5 reports training wall-clock time per environment step and action-inference latency for RAMAC-Diffusion(RADAC), DiffusionQL, and FlowQL under identical hardware and hyperparameters. In these matched settings, RAMAC has comparable (and sometimes slightly lower) wall-clock and latency than the risk-neutral expressive baselines, because the dominant cost comes from the generative actor and evaluating a small number of quantiles for CVaR adds only minor overhead. Flow-based methods also pay their own overhead for training a flow-matching prior and, when used, a distilled one-step policy head, so they are not significantly lighter in practice. Our measurements suggest that such distillation is not necessary in the regimes we study, and we therefore regard it as an interesting optional extension for future work in Appendix A.
> > >
> > >
> > > **(7) OOD detection robustness.**
> > > To check robustness beyond the 1-NN detector, Appendix. E.4 extends the OOD analysis with local outlier factor, and a global Mahalanobis score on the same feature space. Across these detectors and tasks, RAMAC generally shows lower or comparable OOD rates than ORAAC; an exception is a global Mahalanobis score on Hopper-medium-expert, where ORAAC's near-Gaussian collapse around a single mode yields zero estimated OOD mass despite worse hazard and CVaR. We discuss this as a limitation of that global metric and emphasize that local detectors still rank RAMAC as safer.
> > >
> > > We hope these clarifications and new experiments address the concerns and thank the reviewer again for the helpful suggestions.

---

### Official Review · Reviewer_CBrd · 2025-11-01

**Soundness:** 3
**Presentation:** 3
**Contribution:** 3
**Rating:** 6
**Confidence:** 2

**Summary:**

This paper proposes RAMAC (Risk-Aware Multimodal Actor-Critic), a framework for offline reinforcement learning that integrates a generative actor with a distributional critic estimating return quantiles. The method combines behavior cloning to stay close to the data distribution with a CVaR-based risk term that shifts probability mass away from low-return outcomes while maintaining multimodal high-reward behaviors. Theoretical analysis links BC regularization to lower out-of-distribution probability, and experiments on Stochastic-D4RL tasks show improved CVaR performance and competitive mean returns compared to strong baselines such as CQL, CODAC, ORAAC, DiffusionQL, and FlowQL.

**Strengths:**

1. Propagating distributional (CVaR) signals through diffusion/flow trajectories to directly reshape policy mass is an elegant way to combine expressiveness with tail-risk control; this is a nontrivial architectural idea with plausible benefits.
2. The paper identifies an important gap in using expressive generative policies for risk-aware offline reinforcement learning and explains that previous approaches either restricted policy expressiveness or remained risk-neutral. The motivation and taxonomy of prior mechanisms are well organized.
3. The paper identifies an important gap in using expressive generative policies for risk-aware offline reinforcement learning and explains that previous approaches either restricted policy expressiveness or remained risk-neutral. The motivation and taxonomy of prior mechanisms are well organized.

**Weaknesses:**

1. Backpropagating CVaR through a multi-step diffusion or flow trajectory may be computationally expensive; the paper does not quantify training time, memory, or step cost relative to baselines (DiffusionQL / FlowQL) which could matter in practice.
2. The ablation shows alternatives (Wang, CPW) but more analysis is needed on when CVaR is preferable and trade-offs (variance, mean vs tail). The role of K (tail samples) and N (quantiles) on estimator variance could be expanded.

**Questions:**

See weakness part

---

> ### Author Response · Authors · 2025-11-21
>
> We thank the reviewer for the positive assessment of the contributions and for highlighting two important aspects that deserve more discussion: (i) the computational overhead of backpropagating CVaR through a multi-step diffusion/flow trajectory, and (ii) the choice of risk measure and the role of the estimator hyperparameters K (tail samples) and N (quantile particles). We address each in turn.
>
>    1.Computational cost (training time and step cost):
>
> We agree that backpropagating a CVaR objective through a multi-step diffusion or flow trajectory is indeed more expensive than training a risk-neutral critic with a unimodal actor. In our implementation, however, most of this cost is shared with the expressive baselines (DiffusionQL / FlowQL): RAMAC/RAFMAC reuse the same denoising network, diffusion/ODE schedule, and training budget, and the CVaR objective only adds extra forward/backward passes through the distributional critic. The behavior-regularization terms are simple per-state penalties and do not introduce additional network evaluations.
>
> We are currently running a small runtime study on subset of representative Stochastic-D4RL tasks that reports: (i) wall-clock training time for a fixed number of gradient updates under identical hardware and training budgets and (ii) action inference latency (actions per second) comparing RADAC/RAFMAC to DiffusionQL/FlowQL. We will include these results in the revised version to quantify that the additional CVaR and behavior-regularization extensions add overhead relative to the expressive baselines. We also agree that a distilled one-step head is a promising way to further reduce inference latency.  We plan to explore it in follow-up work. If time permits and we obtain stable results for a distilled head on at least one representative task during the discussion phase, we will report them as an additional ablation in the revised manuscript.
>
>
>    2. Risk measure choice and role of K, N:
>
> We agree that our current draft does not clearly explain when CVaR is preferable and how K and N affect estimator variance. Conceptually, we chose CVaR because it is a coherent, law-invariant risk measure that directly targets the lower tail, which is most relevant in the presence of rare but catastrophic penalties. Distortions such as Wang/CPW provide a softer reweighting of the distribution; in our experiments they tend to improve mean performance but under-control the worst-α quantiles compared to CVaR.  We will add this detail in the revision.
>
> We are currently expanding the discussion around Table 5 to explicitly summarize the observed trade-offs between mean and CVaR₀.₁  and we are running a short sensitivity study on a representative Stochastic-D4RL environment that varies K (number of tail samples used in the empirical CVaR) while keeping the number of quantile particles N fixed to the value used throughout the paper. For this study we will report how the estimated mean and CVaR₀.₁, together with their standard errors across seeds, change as K is varied, and we will comment on the observed variance and mean–tail trade-offs. In addition, we will clarify the role of N and justify our fixed choice in the text (discussing the standard variance/bias and computational-cost trade-offs for larger N) so that our CVaR estimator design is more transparent and better grounded empirically.

---

> > ### Author Response · Authors · 2025-12-03
> >
> > We thank the reviewer again for the detailed comments and positive feedbacks. Below we summarize the changes in the revised manuscript and how they address your questions.
> >
> > **Runtime and latency.**
> > In the revised version we added a dedicated paragraph *“Runtime and latency”* in the experiments section and a corresponding figure in Appendix E.5 (Runtime/latency study). There we report, on a representative Stochastic-D4RL control task, both (i) wall-clock training time under identical hardware and training budgets and (ii) per-action inference latency for RAMAC (RADAC), DiffusionQL, and FlowQL. As anticipated, most of the cost is shared with the expressive baselines: the dominant component is the diffusion/flow denoising trajectory, while the additional IQN/CVaR evaluations introduce only a modest overhead. Empirically, RAMAC remains in the same practical regime of training time and inference speed as DiffusionQL/FlowQL, with differences smaller than typical variance across hyperparameter choices.
> >
> > **Estimator design and roles of (K) and (N).**
> > To address the question about the CVaR estimator, we have added explanation in a new appendix subsection E.6 that the exact estimator used in our implementation: RAMAC uses an IQN critic with (N) quantile particles to approximate the return distribution and a Monte Carlo estimator with (K) samples from $\tau \sim \mathcal{U}(0,\alpha)$ to estimate $\mathrm{CVaR}_\alpha$. We discuss the standard bias–variance and computational trade-offs when increasing (K) and (N).
> >
> > In addition, we added an empirical study in Appendix E.6 (“CVaR estimator variance vs.\ $K$’’) on two representative Stochastic-D4RL tasks, \textsc{halfcheetah-medium-replay-v2} and \textsc{walker2d-medium-replay-v2}. For a fixed $N$, we measure the empirical variance of the IQN-based $\mathrm{CVaR}_\alpha$ estimator as a function of $K \in \{2,4,8,16\}$ and $\alpha \in \{0.05,0.1,0.2\}$ over many repetitions and evaluation states. The plots show that the variance decreases rapidly between $K=2$ and $K=4/8$, with diminishing returns beyond $K=8$, and that the three $\alpha$-levels produce very similar curves once $K\ge 4$. This supports our choice of a moderate $K$, which substantially stabilizes the estimator without incurring prohibitive extra computation, while treating $N$ as a fixed architectural hyperparameter.
> >
> > We hope these additions make the runtime behavior, and the CVaR estimator design (including the roles of (K) and (N)) more transparent.
> >
> > Please let us know if there is anything else we can clarify

---

### Author Response · Authors · 2025-12-03
**Summary of rebuttal and revisions (for AC)**

Dear Area Chair,

Below we briefly summarize the main concerns of each reviewer and how the revision addresses them.

**Reviewer CBrd.**

**Main concerns:** computational overhead of backpropagating a CVaR objective through diffusion/flow trajectories, and limited discussion of risk-measure choice and the roles of K (tail samples) and N (quantile particles).

**In the revised manuscript,** we added a “Runtime and latency” paragraph and a figure reporting wall-clock training time and per-action inference latency for RADAC vs. DiffusionQL/FlowQL under matched hardware, showing that the extra IQN/CVaR evaluations incur only modest overhead relative to the generative actor. We also added a subsection that details the IQN-based CVaR estimator, explains the bias–variance and cost trade-offs in K and N, includes a variance study supporting our moderate K choice, and clarifies why we use CVaR as the default distortion versus Wang/CPW.

**Reviewer JUg6.**

**Main concerns:** need for a clearer distinction from concurrent “distributional critic + diffusion/flow” methods that still optimize expected return, missing related work (Luo et al.), clarification of hazard relabeling and potential label-leakage risks, explicit risk-aware baselines (CODAC-CVaR, CQL-CVaR, SPIBB-style), a risk–return frontier over α, runtime/latency, and OOD detection robustness.

**In the revised manuscript,** Sections 1–3 emphasize that RAMAC trains a single expressive generative actor directly on a BC+CVaR objective and uses the BC (forward-KL) term to bound per-state OOD mass, in contrast to expectation-based updates in prior diffusion/flow work; we also add Luo et al. to related work. We describe the hazard relabeling procedure and clarify that our CODAC configuration is already CVaR-optimizing and discuss why a “CQL-CVaR” would be conceptually similar, and position SPIBB-style high-confidence methods as complementary future work. We further add a risk–return frontier over α, report runtime/latency measurements, and extend OOD analysis with alternative detectors, where RADAC generally maintains lower or comparable OOD mass than ORAAC.

**Reviewer ktys.**

**Main concerns:** presentation too terse (offline RL, OOD, risk-sensitive RL, distributional RL, behavior regularization, diffusion), RAFMAC feeling like an underdeveloped afterthought, theory claims stronger than what is actually proved, unclear toy bandit and stochastic-D4RL hazard setup, and a need for stronger, more visible ablations.

**In the revised manuscript,** we expanded the preliminaries and early method section to define offline RL and OOD actions, briefly review behavior-regularized methods, introduce distributional critics and CVaR in a self-contained way, and describe diffusion/flow policies as differentiable trajectories; we also explicitly define the BC loss and fix minor wording/typo issues. RAFMAC is now de-emphasized: RADAC (diffusion) is the primary RAMAC instantiation in the main text, while RAFMAC is moved to the appendix as a supplementary flow-matching variant. The theory section is rewritten to focus on two concrete points, a per-state OOD bound for BC-regularized expressive actors via a TV–KL argument, and a geometric analysis of prior-anchored perturbation on thin/non-convex supports, and we clarify the Toy Risky Bandit and hazard construction, add rollout return-distribution plots, and strengthen the ablation story through a OOD-action analysis, toy sweeps, and cross-references to distortion/α/K/N studies.

**Reviewer dseA.**

**Main concerns:** perceived lack of novelty beyond “Diffusion Q-Learning + distributional critic”, questions about the soundness and tuning of the toy experiment and Fig. 3, doubts that Fig. 4 really demonstrates “safety” when RADAC optimizes CVaR of reward, and the sense that the theoretical insights are largely known.

**In the revised manuscript,** the introduction and early sections now make explicit that, RAMAC is the first offline risk-aware framework that (a) couples a diffusion/flow actor with a direct CVaR objective and (b) links its BC regularizer to a per-state OOD bound specialized to expressive generative policies. For the Toy Risky Bandit we fully describe the safe-center / risky-ring geometry, show that BC-only generators recover both modes, and add hyperparameter sweeps over BC/RL weights and α across methods, highlighting structural trade-offs for risk-neutral expressive and anchor-perturbation baselines relative to RADAC. We add the hazard model rollout and return-distribution plots with mean/median/CVaR markers that, together with hazard counts, show that RADAC reduces catastrophic tails while preserving high-return modes, and refocus the theory section on the two specific observations above rather than broad “expressive policies help” statements.

---

### Note · Program_Chairs · 2026-01-17
**Submission Desk Rejected by Program Chairs**

The following references in this submission do not refer to real documents and/or have major errors in bibliographic information:

 Xiao Chen, Bowen Tan, and Pieter Abbeel. Planning with diffusion models. In International Conference on Learning Representations (ICLR), 2023.